# Carrier-phonon decoupling in perovskite thermoelectrics via entropy engineering

Yunpeng Zheng [1,2,12], Qinghua Zhang [3,12], Caijuan Shi[4,12], Zhifang Zhou [1,12], Yang Lu[5], Jian Han[1], Hetian Chen[1], Yunpeng Ma[1], Yujun Zhang [4], Changpeng Lin [6], Wei Xu[4,7], Weigang Ma [5], Qian Li [1], Yueyang Yang[1], Bin Wei [1,8], Bingbing Yang[1,9], Mingchu Zou[1], Wenyu Zhang[1], Chang Liu[1], Lvye Dou[1], Dongliang Yang[4], Jin-Le Lan [10], Di Yi [1], Xing Zhang [5], Lin Gu [11], Ce-Wen Nan [1] & Yuan-Hua Lin [1] ✉

Thermoelectrics converting heat and electricity directly attract broad attentions. To enhance the thermoelectric figure of merit, $zT$, one of the key points is to decouple the carrier-phonon transport. Here, we propose an entropy engineering strategy to realize the carrier-phonon decoupling in the typical $SrTiO_3$-based perovskite thermoelectrics. By high-entropy design, the lattice thermal conductivity could be reduced nearly to the amorphous limit, $1.25\,\mathrm{W\,m^{-1}\,K^{-1}}$. Simultaneously, entropy engineering can tune the Ti displacement, improving the weighted mobility to $65\,\mathrm{cm^2\,V^{-1}\,s^{-1}}$. Such carrier-phonon decoupling behaviors enable the greatly enhanced $\mu_W/\kappa_L$ of $\sim 5.2 \times 10^3\,\mathrm{cm^3\,K\,J^{-1}\,V^{-1}}$. The measured maximum $zT$ of 0.24 at 488 K and the estimated $zT$ of ~0.8 at 1173 K in $(Sr_{0.2}Ba_{0.2}Ca_{0.2}Pb_{0.2}La_{0.2})TiO_3$ film are among the best of $n$-type thermoelectric oxides. These results reveal that the entropy engineering may be a promising strategy to decouple the carrier-phonon transport and achieve higher $zT$ in thermoelectrics.

Thermoelectrics converting electricity and heat directly are promising in waste heat harvesting and active cooling, providing solutions to the fossil fuel crisis and 5G/6G micro-device cooling[1]. To quantify the comprehensive thermoelectric properties, the dimensionless figure of merit, $zT = S^2\sigma T/(\kappa_e + \kappa_L)$, was employed, where $S$, $\sigma$, $T$, $\kappa_e$, $\kappa_L$ represent the Seebeck coefficient, electrical conductivity, temperature in Kelvin, carrier thermal conductivity and lattice thermal conductivity, respectively[2]. Accordingly, the simultaneous realization of a high Seebeck coefficient, high electrical conductivity, and low thermal conductivity, makes good thermoelectrics, requiring synergistic modulation of carrier-phonon transport[3].

However, the parameters in the $zT$ definition are strongly intercorrelated. The quality factor $B = \left(\frac{k_B}{e}\right)^2 \frac{8\pi e(2m_e k_B T)^{\frac{3}{2}}}{3h^3} \frac{\mu_W}{\kappa_L} T$, positively correlated to $zT$, indicates that the extent of carrier-phonon decoupling

[1]State Key Laboratory of New Ceramics and Fine Processing, School of Materials Science and Engineering, Tsinghua University, Beijing, PR China. [2]Key Laboratory of Eco-materials Advanced Technology, College of Materials Science and Engineering, Fuzhou University, Fuzhou, PR China. [3]Beijing National Laboratory for Condensed Matter Physics, Institute of Physics, Chinese Academy of Sciences, Beijing, China. [4]Beijing Synchrotron Radiation Facility, Institute of High Energy Physics, Chinese Academy of Sciences, Beijing, China. [5]Key Laboratory for Thermal Science and Power Engineering of Ministry of Education, Department of Engineering Mechanics, Tsinghua University, Beijing, China. [6]Theory and Simulation of Materials (THEOS), École Polytechnique Fédérale de Lausanne, Lausanne, Switzerland. [7]RICMASS, Rome International Center for Materials Science Superstripes, Via dei Sabelli 119A, Roma, Italy. [8]Henan Key Laboratory of Materials on Deep-Earth Engineering, School of Materials Science and Engineering, Henan Polytechnic University, Jiaozuo, PR China. [9]Key Laboratory of Materials Physics Institute of Solid State Physics, Chinese Academy of Sciences, Hefei, China. [10]State Key Laboratory of Organic-inorganic Composite, College of Materials Science and Engineering, Beijing University of Chemical Technology, Beijing, PR China. [11]National Center of Electron Microscopy in Beijing, School of Materials Science and Engineering, Tsinghua University, Beijing, China. [12]These authors contributed equally: Yunpeng Zheng, Qinghua Zhang, Caijuan Shi, Zhifang Zhou. ✉e-mail: linyh@tsinghua.edu.cn

reflected by $\frac{\mu_w}{\kappa_L}$ is the key point to the $zT$ enhancement[4]. Recently, many efforts have been made to decouple carrier-phonon transport. By modulation of intrinsic crystal symmetry, e.g., *Pnma* to *Cmcm* symmetry in SnSe by Pb and Cl co-doping[5], rhombohedral to near cubic phase transition in Pb and Sb co-doped GeTe[6], the resulting high symmetry could benefit the improvement of the Hall mobility, while the thermal conductivity was suppressed by the dopants. Additionally, through compositing effects by adding metal particles[7], band aligning precipitates[4,8] in the matrix phase, energy barriers for carrier transport were lowered, and the phonons were simultaneously scattered. Furthermore, interface engineering, e.g., constructing PbTe-SrTe coherent interfaces in PbTe system[9], forming $Cu_2Se$-BiCuSeO-graphene interfaces in $Cu_2Se$ materials[10], was also proved effective to concurrently achieve phonon blocking and charge transmitting[9]. Despite such efforts in alloys to decouple carriers and phonons effectively, carrier-phonon decoupling is still a big challenge in thermoelectric oxides.

Oxides of low cost, low pollution, high abundance, and excellent thermal stability are competitive in thermoelectrics[11]. $SrTiO_3$, as a typical perovskite thermoelectric oxide, displays relatively good electrical properties[12] (PF = $S^2\sigma$ over 1000 $\mu W\,m^{-1}\,K^{-2}$) originating from special $TiO_6$ octahedrons and high symmetry cubic phase[13], but suffers from high thermal conductivity over 11 $W\,m^{-1}\,K^{-1}$[14]. Strategies such as nanostructuring[15], element doping[16], and vacancy modulation[17], were applied in reducing lattice thermal conductivity, but the carrier mobility was sacrificed, limiting the further enhancement of $zT$ in $SrTiO_3$-based thermoelectrics[18]. Therefore, carrier-phonon decoupling is important for $zT$ enhancement in $SrTiO_3$-based oxides. Modifying the grain boundaries of $SrTiO_3$-based ceramics with carbon-based materials could synergistically tune the electrical-thermal transport from the extrinsic compositing aspect[19-24]. However, it is not quite common to decouple carrier-phonon transport in $SrTiO_3$ by intrinsically tuning the composition and structure.

Recently, the high-entropy strategy has been found to remarkably suppress the lattice thermal conductivity in thermal barrier coatings and thermoelectrics[25-27]. Normally, ion displacement is strongly related to carrier transport[28], which could be regulated in perovskites-based dielectric capacitors by engineering entropy[29,30]. Herein, we propose that rational entropy engineering could tune the phonon and carrier transport behaviors, which could be expected to decouple the carrier-phonon transport, reaching an overall optimization of the thermoelectric performance in $SrTiO_3$-based oxides.

## Results

### Design and preparation of entropy-engineered thermoelectrics

As for the crystal sites for entropy engineering, since the Ti 3*d* orbitals form the conduction band minimum (CBM) deciding the electron transport in *n*-type $SrTiO_3$-based semiconductors[31], and A-site vibrations correspond to acoustic phonon branches and low-frequency optical branches dominating lattice thermal conductivity[32,33], the A-site entropy engineering could help maintain the $TiO_6$ octahedrons undisturbed to prevent mobility deterioration and meanwhile strongly scatter phonons[34-36], realizing carrier-phonon decoupling. Therefore, $SrTiO_3$, $BaTiO_3$, $CaTiO_3$, and $PbTiO_3$ were selected to form solid solutions. Starting from La-doped $SrTiO_3$ and keeping the doping level (20%) of La fixed, the entropy was engineered by introducing Ba, Ca, and Pb, equimolarly occupying the rest 80% of A sites. $(Sr_{0.8}La_{0.2})TiO_3$ (SLTO), $(Sr_{0.4}Ba_{0.4}La_{0.2})TiO_3$ (SBLTO), $(Sr_{0.267}Ba_{0.267}Ca_{0.267}La_{0.2})TiO_3$ (SBCLTO), and $(Sr_{0.2}Ba_{0.2}Ca_{0.2}Pb_{0.2}La_{0.2})TiO_3$ (SBCPLTO) films were grown on single crystal $(LaAlO_3)_{0.3}(Sr_2TaAlO_6)_{0.7}$ (LSAT) (001) substrates by pulsed laser deposition (PLD) followed by annealing in reducing atmosphere to create O vacancies and provide electrons to make the films conductors (see below in "Methods" section). The reason for fabricating the thin films, rather than bulks, is that the high-quality thin films could reduce the influence of grain boundaries in perovskites on electrical and thermal transport[37,38], making it easier to

construct entropy-transport relationship and enhance the thermoelectric properties. According to the definition of the configuration entropy $S_{config.} = -R((\sum_{i=1}^{N} x_i \ln x_i)_{cation-site} + (\sum_{j=1}^{M} x_j \ln x_j)_{anion-site})$, where $R$, $N$, $M$, $x$ are ideal gas constant, the number of cation-site elements, the number of anion-site elements, and the molar ratio of elements[39], the $S_{config.}$ increased from low entropy (0.50$R$ in SLTO) to medium entropy (1.05$R$, 1.38$R$ in SBLTO, SBCLTO, respectively), and to high entropy (1.61$R$ in SBCPLTO without taking O and Pb vacancies into account) (Supplementary Table 1) in this work.

The XRD patterns (Supplementary Figs. 1–3) showed that perovskite thin films were epitaxially grown of high quality by PLD on LSAT (001) substrates, and there were no impure phases before and after the annealing process. The unchanged position and shape of LSAT peaks (Supplementary Fig. 3) indicated chemical endurance to reduce, avoiding electrical contribution from substrates[40]. The shifts after sequent introduction of Ba, Ca, Pb to $(Sr_{0.8}La_{0.2})TiO_3$ were consistent with the relative size changes of introduced A-site atoms (Supplementary Fig. 1, Supplementary Table 2), illustrating the successful solid solution at A sites. The epitaxy nature could also be revealed by RSM of (103) peaks (Supplementary Fig. 4), and the peaks of the films deviated from the $Q_x$ of LSAT (103), which was a sign for in-plane strain release by thickness of around 200 nm, excluding the influence which epitaxial strain could make on transport. The interface of the thin films and substrates consisted of co-vertex connected Al/$TaO_6$ and $TiO_6$ octahedrons through ABF and HAADF images (Supplementary Figs. 5 and 9h), and the films showed good morphology in TEM (Supplementary Fig. 6), verifying coherently epitaxial thin films of high quality. The elements distributed uniformly at A sites in low-entropy, medium-entropy, and high-entropy samples, and no obvious ordered structure and distribution were viewed, which was reflected by EDS mapping in atomic resolution and micro-scale (Supplementary Figs. 7–11), confirming the random and homogeneous occupation at A sites to fulfill the definition of high entropy. Unavoidable Pb volatilization during target sintering, film deposition, and annealing contributed to the considerable amount of Pb vacancies (Supplementary Table 3, the molar ratio Pb/Ti = 0.073 by EPMA, nominal $V_{Pb}$/Ti = 0.127), adding to the entropy at A sites and tuning the A-site average radius.

### Suppressing lattice thermal conductivity to amorphous limit

Since A-O skeleton dominates the transport of acoustic and low-frequency optical phonons[17], high level of La doping (20%) in SLTO primarily reduced lattice thermal conductivity from 11 $W\,m^{-1}\,K^{-1}$ of STO single crystals[12] to 2.52 $W\,m^{-1}\,K^{-1}$ due to size, mass, charge differences and the vacancies introduced (Fig. 1). The entropy engineering at A sites could further strongly suppress the lattice thermal conductivity $\kappa_L$, and thermal conductivity of ~1.60 $W\,m^{-1}\,K^{-1}$ at room temperature was measured by TDTR in SBCPLTO (Fig. 1a and Supplementary Fig. 26). After increasing entropy by introducing elements, the phonon mean free path $l_p$ decreased monotonically (Supplementary Table 7), and the room temperature lattice thermal conductivity of thin films was suppressed significantly from 2.52 $W\,m^{-1}\,K^{-1}$ for SLTO to 1.25 $W\,m^{-1}\,K^{-1}$ for SBCPLTO, approaching the amorphous limit[41] (Fig. 1b, Supplementary Fig. 12). To verify the thermal measurement of thin films, the lattice thermal conductivity of corresponding bulks was measured by LFA, and the results between TDTR and LFA, bulks and films, were close to some extent in Supplementary Fig. 24.

Intrinsic structural factors and extrinsic defect factors could explain the entropy-scattering relations. The acoustic and low-frequency optical branches of phonons are mainly influenced by A sites in perovskites[32], and thus for intrinsic factors, the doping at A sites with different mass, size, charge and vacancies could effectively scatter phonons[25,42]. The size disorder, described by a standard deviation, size disorder parameter $\delta$ (Supplementary Equation 1), could lead to lattice

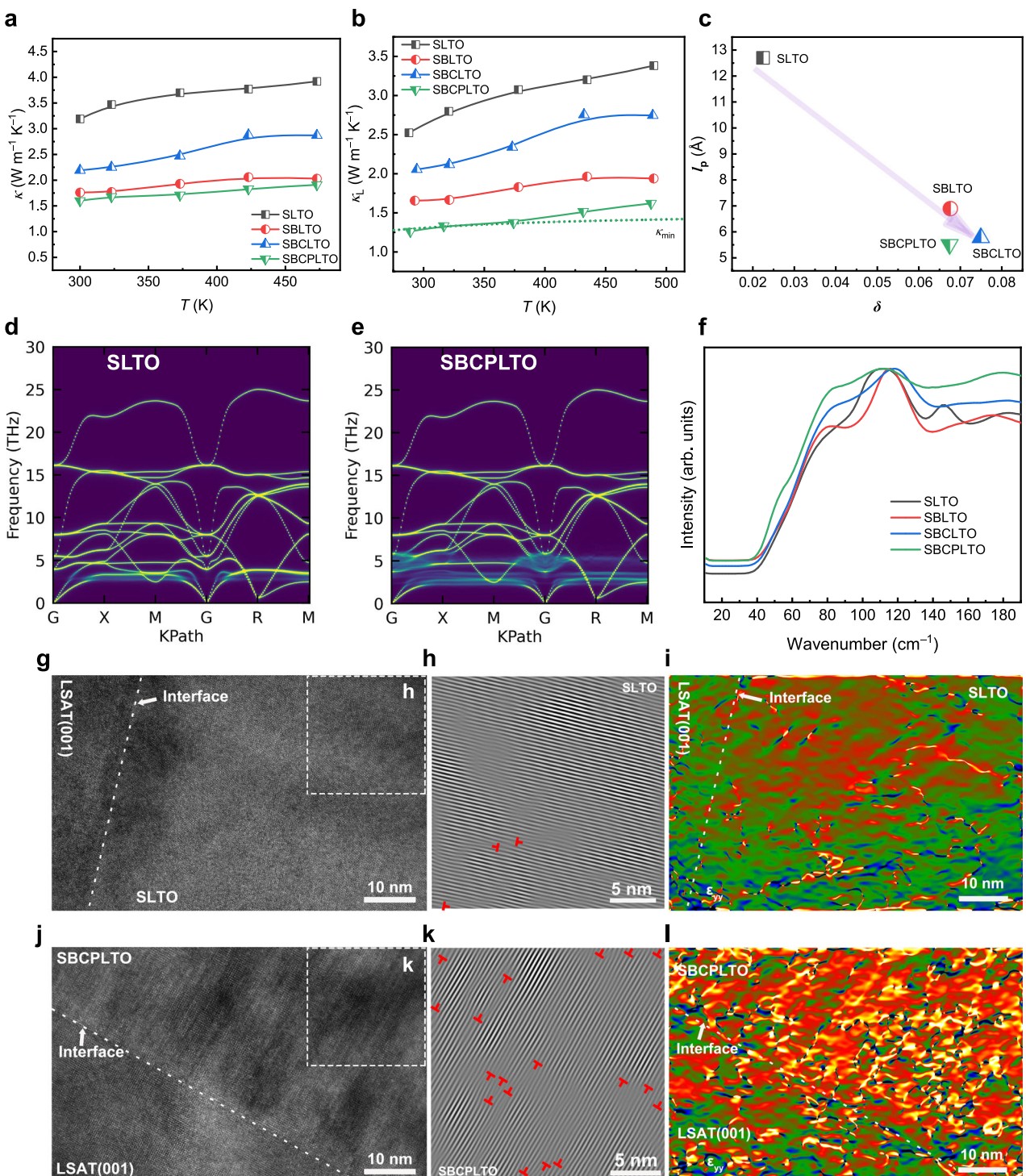

**Fig. 1 | Thermal transport behaviors of entropy-engineered thin films, and the extrinsic and intrinsic origins of strong phonon scattering after entropy increase. a** Temperature-dependent total thermal conductivity ($\kappa$) of entropy-engineered perovskite oxide thin films. **b** Calculated lattice thermal conductivity ($\kappa_L$) of entropy-engineered thin films and amorphous limit thermal conductivity of SBCPLTO. **c** The relation between phonon mean free path ($l_p$) and the size disorder parameter ($\delta$) (Supplementary Table 4)[43,99] at A sites. The phonon mean free path was measured on corresponding bulks at room temperature for reference

(Supplementary Table 7). The definition of size disorder parameter $\delta$ can be found in supplementary materials (Supplementary Equation 1). The purple arrow is a guide for the eyes. **d**, **e** The DFT calculation of phonon dispersion of SLTO (**d**) and SBCPLTO (**e**). **f** Raman spectra of corresponding entropy-engineered bulks for reference. **g**–**l** The TEM, FFT, and GPA results of SLTO (**g**, **h**, **i**) and SBCPLTO (**j**, **k**, **l**). The dashed lines mark the interfaces between the films and substrates. The dislocation symbols are also shown in red.

distortion, which is thought to be the main factor suppressing lattice thermal conductivity[43]. Due to the comparably large radius of $Ba^{2+}$ (Supplementary Table 2), the significantly increased size disorder had already strongly scattered phonons and lowered the $l_p$ (Fig. 1c). In

addition to the size disorder coming from different radius of elements, the charge difference from $La^{3+}$ also could be seen as one kind of size fluctuation. Causing lattice distortion by the compensation for the local charge imbalance[25], the charge disorder is usually responsible for

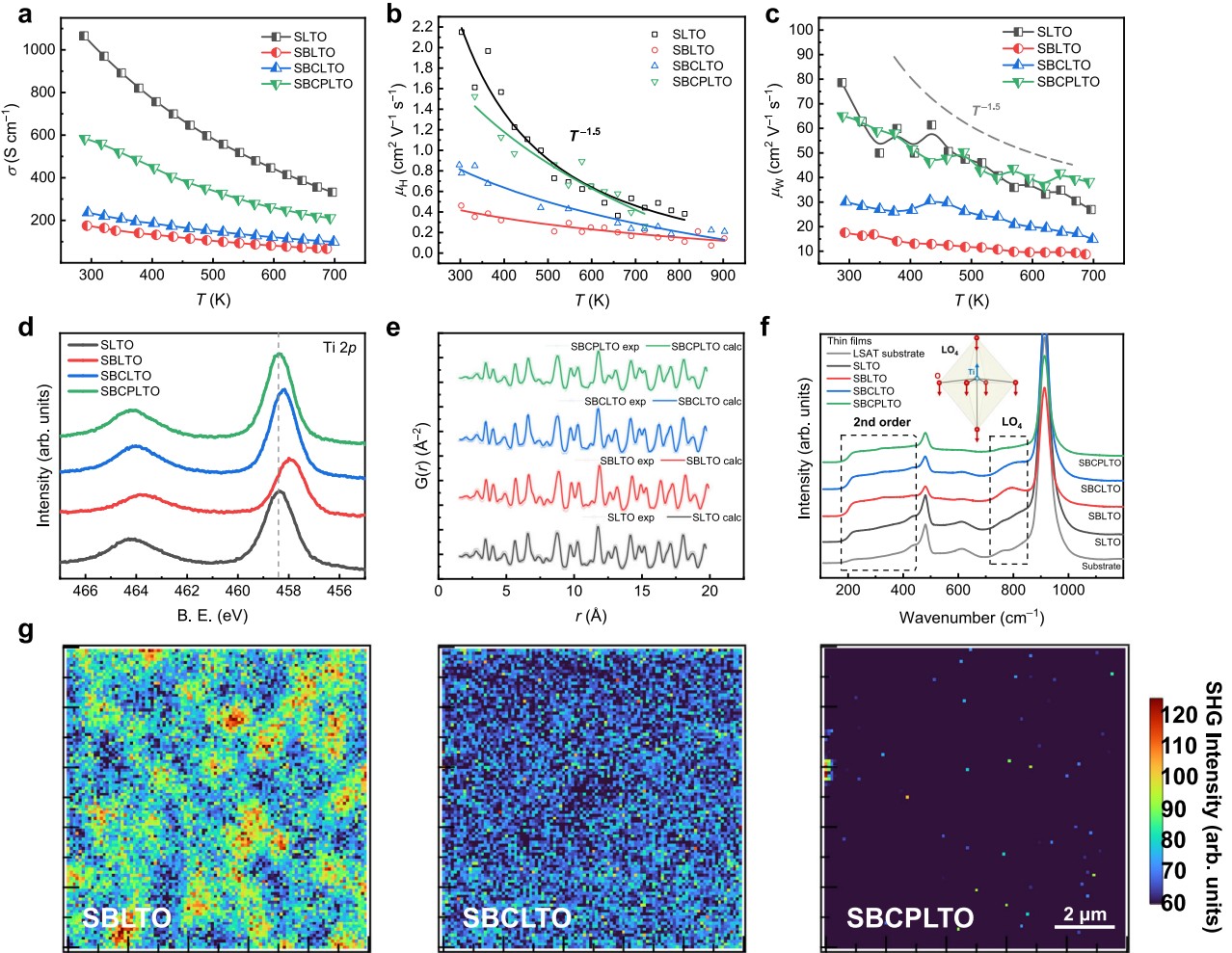

**Fig. 2 | Electrical transport behaviors of entropy-engineered thin films and the possible structural origins of decoupled carrier transport explained by spectral methods. a–c** Temperature-dependent electrical conductivity ($\sigma$) (**a**), Hall mobility ($\mu_H$) (**b**), and weighted mobility ($\mu_W$) (**c**) of entropy-engineered thin films. The acoustic phonon scattering dominated $T^{-1.5}$ trend curves were plotted (dashed gray line). **d** The XPS Ti $2p$ spectra of entropy-engineered thin films. The dashed gray line marks the peak position and serves as a guide for the eyes. **e** The PDF results of corresponding bulks to get referencing bond length and the fitting curves by reverse Monte Carlo methods. **f** The Raman spectra of LSAT substrates and entropy-engineered thin films on LSAT substrates. The typical vibration modes of perovskite thin films were highlighted by dashed lines, and the inset is the schematic of $LO_4$ vibration mode. **g** SHG mapping of SBLTO, SBCLTO, and SBCPLTO in $10\,\mu m \times 10\,\mu m$.

the strong reduction in thermal conductivity[16,33]. The mass disorder could contribute to phonon scattering as well. As suggested by the phonon dispersion calculation in Fig. 1d, e and Supplementary Fig. 13, by introducing mass disorder to $SrTiO_3$, the acoustic phonon branches, and low-frequency optical phonon branches became broader in dispersion, meaning the enhanced scattering rate of phonons[33]. Furthermore, since the A sites were nominally fully occupied, and the $La^{3+}$, $Sr^{2+}$, $Ba^{2+}$, $Ca^{2+}$, and $Pb^{2+}$ have no other smaller valence states when performing pulsed laser deposition, the A-site vacancies tended to form for charge balance in low-pressure, possibly contributing to the reduction in thermal conductivity by large size, charge and mass disorder[16,19,33]. Consistent with the size disorder parameter $\delta$ and DFT calculation (considering mass disorder), in corresponding unannealed bulks, the Raman spectra (Fig. 1f) which are the projection of optical phonon branches at zero point (G) of the Brillouin zone, displayed broadened peaks around $100\,cm^{-1}$ from SLTO to SBCPLTO, corresponding to increased linewidth of optical branches decided by A-site vibrations[44]. For extrinsic defects, as illustrated by FFT and GPA process from HRTEM images (Fig. 1g–l), larger amounts of dislocations (as reported in high-entropy thermal barriers[27]) and denser strain distribution, were found in high-entropy SBCPLTO than those in SLTO

with lower entropy, functioning as phonon scatters of different scales. In addition, it is worth noting that though the entropy of SBCLTO is higher than SBLTO, the thermal capacity was not necessarily monotonically correlated with entropy, and the Ca with smaller mass could lead to larger thermal capacity (Dulong–Petit Law and Neumann–Kopp Rule[45]), thus larger thermal conductivity in SBCLTO than that in SBLTO (Fig. 1b and Supplementary Fig. 24).

## Improving the carrier mobility by tuning electron-conducting $TiO_6$ octahedrons

The carrier transport including electrical conductivity $\sigma$, Hall mobility $\mu_H$, and weighted mobility $\mu_W$[46] was measured and calculated (Fig. 2a–c and Supplementary Fig. 27). As the carrier concentration was in the order of $\sim 10^{21}\,cm^{-3}$ for the films (Supplementary Table 8), far larger than the critical concentration of degeneration[47], all the films were degenerate semiconductors. For the absence of grain boundary scattering in epitaxial thin films, the $\sigma$-$T$, $\mu_H$-$T$, and $\mu_W$-$T$ followed the acoustic phonon scattering dominated $T^{-1.5}$ trend[48]. SLTO, as a typical thermoelectric material, displayed $\sigma$ of $\sim 1066\,S\,cm^{-1}$ and $\mu_H$ over $2.1\,cm^2\,V^{-1}\,s^{-1}$ at room temperature, while SBLTO, after the introduction of Ba with a much larger radius of $1.61\,Å$, showed deteriorated $\sigma$ of

~175 S cm$^{-1}$ and $\mu_H$ of ~0.4 cm$^2$ V$^{-1}$ s$^{-1}$ at room temperature. Though the lattice thermal conductivity was low enough in SBLTO, but the coupled poor electrical properties harmed the thermoelectric performances. Substituting Ba by Ca, and Pb and simultaneously introducing Pb vacancies with lower radius, the weighted mobility recovered from SBLTO to SBCLTO to SBCPLTO, and SBCPLTO was comparable with SLTO in weighted mobility. The carrier mean free paths $l_c$[48] and the deformation potentials $\Xi_{def}$[5] were calculated (Supplementary Table 8), and, obviously, with decreased deformation potentials, SBCPLTO obtained increased $l_c$. The structural factors behind the recovery of carrier mobility needed exploring.

The relation between the mobility deterioration and the TiO$_6$ distortion was noticed. In Ti 2$p$ XPS (Fig. 2d), there were no obvious Ti$^{3+}$ shoulder peaks near Ti$^{4+}$ in all films[49], suggesting that Ti mainly existed as Ti$^{4+}$. In this case, the shifts of Ti and O XPS binding energy peaks (Fig. 2d and Supplementary Fig. 16) were negatively correlated to the bond length. In terms of lattice parameters, SBLTO > SBCPLTO > SBCLTO > SLTO (Supplementary Fig. 14), while according to XPS, for Ti-O bond length, SBLTO > SBCPLTO > SBCPLTO ≈ SLTO (Fig. 2d and Supplementary Fig. 16). SBCLTO had longer Ti-O bonds though smaller lattice than SBCPLTO, indicating that the TiO$_6$ octahedrons were distorted in SBCLTO, and SBCPLTO had similar Ti-O chemical surroundings with centrosymmetric SLTO despite different lattice parameters. PDF (pair distribution function) by synchrotron radiation is a good method to get information on coordination and bonds. The reverse Monte Carlo simulation could give the average bond length of atom pairs by fitting the pair distribution function (Fig. 2e), and the observed tolerance factors $t_{obs} = \frac{length(A-O)}{\sqrt{2}length(B-O)}$ were calculated (Supplementary Table 6)[13]. The deviation of $t_{obs}$ from 1 in SBLTO and SBCLTO could represent lowered symmetry and distorted TiO$_6$ octahedrons.

Furthermore, the types of TiO$_6$ distortion could also be inferred by spectral methods. In Raman spectra of films (Fig. 2f), compared with substrates, extra peaks from 240 to 400 cm$^{-1}$ in films were considered to be multiple second-order peaks typical in perovskites, and extra peaks around 795 cm$^{-1}$ in SBLTO and SBCLTO corresponded to the LO$_4$ mode, indicating off-center vibration of Ti in TiO$_6$ octahedrons[50]. The absence of LO$_4$ mode in SLTO and SBCPLTO could be thought of as a sign of Ti centrosymmetric vibrations, and the red-shifted and stronger LO$_4$ signal in SBLTO than that in SBCLTO might be from larger displacement and longer bond length. Additionally, from the second harmonic generation (SHG) mapping (Fig. 2g and Supplementary Fig. 17), the breaking of centrosymmetry by displacement could also be inferred. The polar domains were visualized in SBLTO, and the signal weakened in SBCLTO, hence hardly detectable in SBCPLTO with recovered mobility. The Ti displacement seemed to be the type of octahedron distortion accounting for the mobility deterioration.

To validate the TiO$_6$ distortion type, the Cs-corrected STEM was performed. The anti-direction tilting could double the unit cells, and lead to half-index diffraction in the selected area electron diffraction (SAED)[51]. In SLTO and SBCLTO (Supplementary Fig. 18), the presence of half-index diffraction points in SAED patterns along [110] direction ensured the tilting of TiO$_6$, while no half-index diffraction points were obtained in SBLTO and SBCPLTO (Fig. 3b, c). The octahedron tilting could be seen in atomic resolution annular bright field (ABF) images by the displacement of oxygen ion (Supplementary Fig. 19), and the tilting in SBCLTO was stronger than that in SLTO, explaining the smaller binding energy in Ti 2$p$ XPS of SBCLTO. In terms of B-site displacement, the displacement of Ti ions decreased from SBLTO to SBCLTO, and Ti ions hardly displaced in SBCPLTO (Fig. 3d–f). In short, the distortion types were slight TiO$_6$ tilting in SLTO, pure large Ti displacement in SBLTO, mixed TiO$_6$ tilting and moderate Ti displacement in SBCLTO, and minimized distortion in SBCPLTO, respectively (Supplementary Table 10). As can be seen, the TiO$_6$ octahedrons in SLTO

are tilted, and in SBCPLTO, the tilting is ignorable, but the mobility was close between SLTO and SBCPLTO. Furthermore, the TiO$_6$ tilting is ignorable in SBLTO, but the mobility deteriorates significantly, which seems that the tilting is not the main factor affecting carrier mobility. The Ti displacement dominated the electron transport in our entropy-engineered perovskite titanates, and with an increasing displacement of Ti, the weighted mobility decreased (Fig. 3d–g). To explain, the displaced ions could create shorter and longer bonds, and the electrons could be localized in shorter bonds with overlapping electron clouds[28] (Fig. 3a). From the aspect of orbitals, the breaking of the symmetry by Ti displacement could result in irregular bond lengths and O-Ti-O angles departing from 180°, and thus the split of $t_{2g}$ orbitals would lead to carrier localization[13,52,53]. Furthermore, the inhomogeneous displacement among adjacent octahedrons in disordered medium-entropy samples could contribute to inhomogeneous electronic potentials, thus scattering electrons. After improving the symmetry, the similar bond feature could enable homogeneous electron cloud overlapping among Ti-O bonds, delocalizing the electron to transport. In summary, the Ti displacement was the distortion that hindered the electron transport in this case.

## Decoupled carrier-phonon transport and enhanced thermoelectric properties

The discussion on the carrier-phonon decoupling is needed. For phonon transport, the low-entropy SLTO displayed relatively high thermal conductivity (Fig. 1a, b), and the medium-entropy SBLTO, SBCLTO, the high-entropy SBCPLTO showed decreased phonon mean free path $l_p$ and minimized thermal conductivity (Fig. 4a and Supplementary Table 7). The increase of configuration entropy ($S_{config.}$) would lead to the increase in size disorder, mass disorder, and dislocation density[27], and it could be summarized that the thermal diffusivity $D$ decreased with increasing entropy, as shown in Fig. 4b. In terms of carrier transport, after the introduction of large Ba at A sites, the weighted mobility decreased. Interestingly, the weighted mobility was improved with suppressed phonon mean free path $l_p$ from SBLTO to SBCLTO to SBCPLTO by a decreased amount of large Ba cations and substitution by smaller Ca, Pb, and Pb vacancies (Fig. 4a). As discussed above, the displacement of Ti in TiO$_6$ octahedrons was the main factor affecting the carrier transport. The TiO$_6$ octahedrons could be tuned by changing the A-O average bond length, and the tolerance factor $t = \frac{length(A-O)}{\sqrt{2}length(B-O)}$ was the reason behind[13]. When the tolerance factor is 1, the perovskite tends to form a cubic phase with undistorted centrosymmetric BO$_6$ octahedrons, and SrTiO$_3$ is a typical material with $t \approx 1$[53]. The occupation of large Ba at A sites would deviate the $t$ from 1, distorting the TiO$_6$, and the introduction of smaller ions or vacancies to tailor $t$ back to 1 could realize TiO$_6$ recovery. As concluded in Fig. 4c, with the $t_{obs}$ approaching 1, the TiO$_6$ became less distorted, performing better in weighted mobility. Since entropy is a concept of disorder, and the tolerance factor $t$ is a mean value, it is reasonable to decouple the transport behaviors by these two different math concepts. Additionally, the entropy design was at A sites, dominating phonon transport, while the carrier transport was mainly affected by TiO$_6$ octahedrons, which was also the reason for the carrier-phonon decoupling. In summary, A-site rational entropy engineering resulted in A-site disorder, reducing lattice thermal conductivity, and meanwhile modulated the A-site radius and tolerance factor, thus reducing the Ti displacement and improving carrier mobility. Consequently, the carrier-phonon transport was decoupled in perovskite thermoelectrics via entropy engineering. Due to the decoupled carrier-phonon transport, the $\mu_W/\kappa_L$ increased from SBLTO to SBCPLTO, and achieved enhanced $\mu_W/\kappa_L$ of ~5.2 × 10$^3$ cm$^3$ K J$^{-1}$ V$^{-1}$ at room temperature in SBCPLTO, outperforming the original SLTO (Fig. 4d).

As degenerate semiconductors, the Seebeck coefficients ($S$) of all samples increased linearly with temperature (Supplementary Fig. 23a). High $S$ of −228 µV K$^{-1}$ was reached in SBCPLTO at 693 K, and through

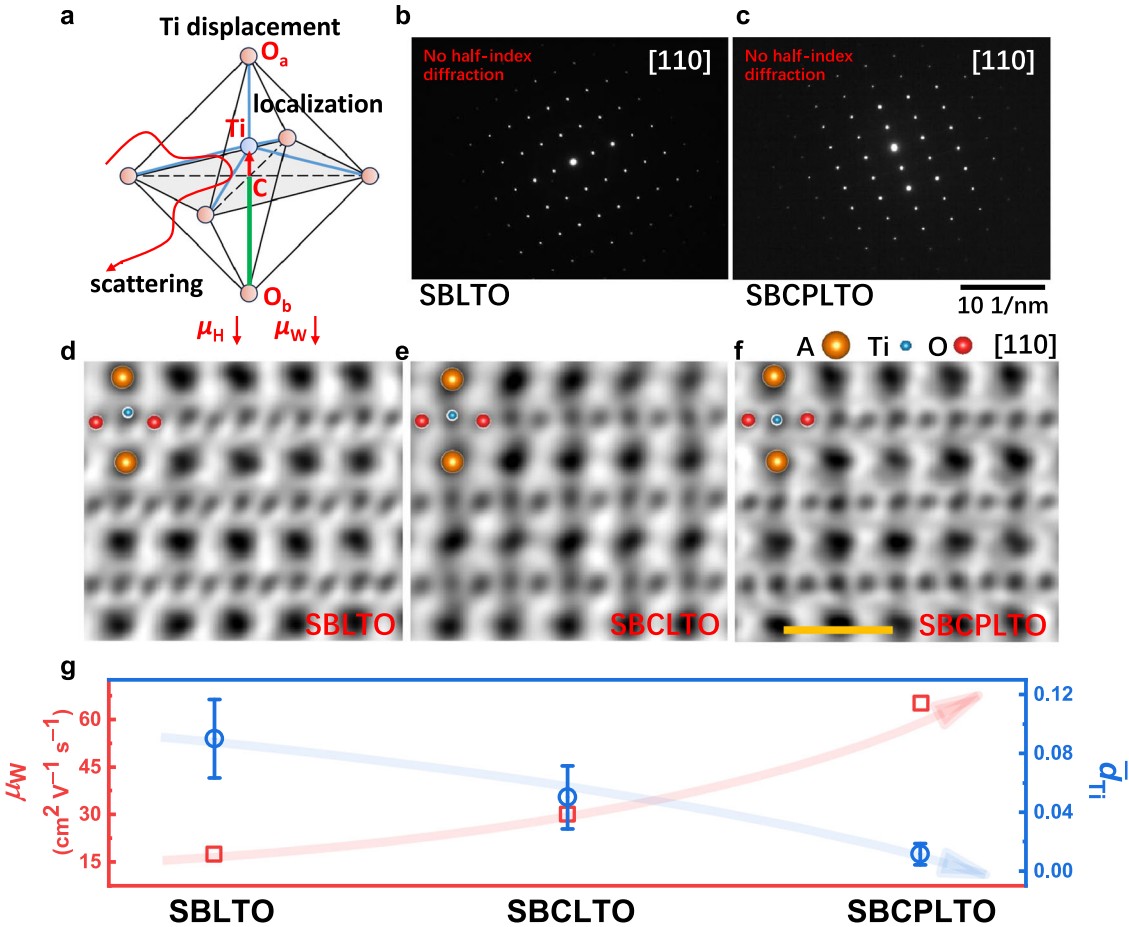

**Fig. 3 | Atomic scale electron microscopy characterization to explain the recovered mobility of entropy-engineered thin films. a** Schematic of the possible mechanism of electron scattering and localization in Ti-displaced $TiO_6$ octahedrons. C point marked is the center of the $TiO_6$ octahedron, and the length of Ti displacement is the length of C-Ti. **b, c** SAED patterns along [110] to prove the absence of octahedron tilting in SBLTO and SBCPLTO. **d–f** ABF images in the atomic resolution after filtering to show the Ti displacement in SBLTO (**d**), SBCLTO (**e**), and SBCPLTO (**f**). The orange scale bar denotes 5 Å. The orange, red, and blue spheres represent A-site atoms, O, and Ti, and the overlapped spheres were omitted. **g** The relation between room temperature weighted mobility ($\mu_W$) and average normalized Ti displacement ($\bar{d}_{Ti}$) of SBLTO, SBCLTO, and SBCPLTO with increased entropy. The displacement was normalized by the ratio $d_{Ti}$ = length(C-Ti)/length(C-$O_b$) in (**a**). The error bar is the standard deviation $\delta_d$. The blue and red arrows are the guides to show the trends of weighted mobility and displacement. The details of the analysis and results can be found in supplementary materials.

Pisarenko plot at room temperature using SPB model (Supplementary Fig. 23b), it could be seen that the $m_d^*$ of SLTO, SBLTO and SBCLTO were close (~6.5 $m_0$), while enhanced $m_d^*$ of ~8.2 $m_0$ was calculated in SBCPLTO. The A-site doping could help tune the energy band structure and the electrical transport in $SrTiO_3$-based thermoelectrics[54,55]. To explain the improved $m_d^*$, the band dispersion was calculated by DFT (Supplementary Fig. 21), and SLTO, SBLTO, and SBCLTO shared similar band structures, while extra orbitals lay at the CBM of SBCPLTO. According to the partial DOS and COHP (Supplementary Figs. 22, 23c, and 29), since Sr, Ba, and Ca belong to the same alkaline-earth main group, with empty $s$ and $d$ orbitals of high energy, the CBM and VBM only consist of Ti-O bonding and anti-bonding orbitals. However, $Pb^{2+}$ with lone pair $6s^2$ and empty $6p$ could affect band structure significantly[54], and coupled with Ti $3d$ and A-site $d$ orbitals, the Pb $6p$ orbitals with high degeneracy could effectively contribute to the DOS at CBM to enhance the $m_d^*$ of SBCPLTO. Additionally, the increased symmetry could also increase the degeneracy, thus the $m_d^*$ in SBCPLTO[56]. Benefiting from recovered mobility and enhanced effective mass, the PF and electronic quality factor $B_E^{57}$ of SBCPLTO outperformed the base material SLTO, and PF reached $1.1 \times 10^3$ μW m$^{-1}$ K$^{-2}$ in SBCPLTO (Fig. 4e and Supplementary Fig. 23d). In SBCPLTO, the decoupled $\mu_W/\kappa_L$ (Fig. 4f) and the increased effective mass combined helped significantly enhance the $zT$ compared to low- and medium-

entropy samples (Supplementary Fig. 23e), and the maximum $zT$ of 0.24 was achieved at 488 K in SBCPLTO (limited by the TDTR measuring temperature). Considering the PF-$T$, $\kappa_L$-$T$ plateau at the middle to high-temperature range in SBCPLTO, the $zT$ was roughly extrapolated linearly to 1173 K, reaching a high estimated $zT$ over 0.8, which outperforms other $n$-type oxide thermoelectrics (Fig. 4g)[12,58–62]. Due to the absence of grain boundary scattering in epitaxial thin films to enhance room temperature performances and the synergistic entropy engineering to decouple carrier-phonon transport, the average $zT$ (RT to 473 K) of 0.14 was obtained (Supplementary Fig. 23f), which is also competitive in $n$-type thermoelectric oxides.

## Discussion

Entropy increase usually accompanies with deteriorated carrier mobility despite lattice thermal conductivity reduction. By applying entropy engineering at phonon transport skeletons and intentionally tuning the relative sizes of ions under the guidance of factors like tolerance factor to delocalize carriers, the carrier-phonon decoupling could be realized. The strategy could be used in thermoelectrics with separate carrier and phonon transport units. Since the high entropy could suppress lattice thermal conductivity to the amorphous limit, the decoupled carrier transport could further enable competitive thermoelectric properties. This work provides solutions to applying

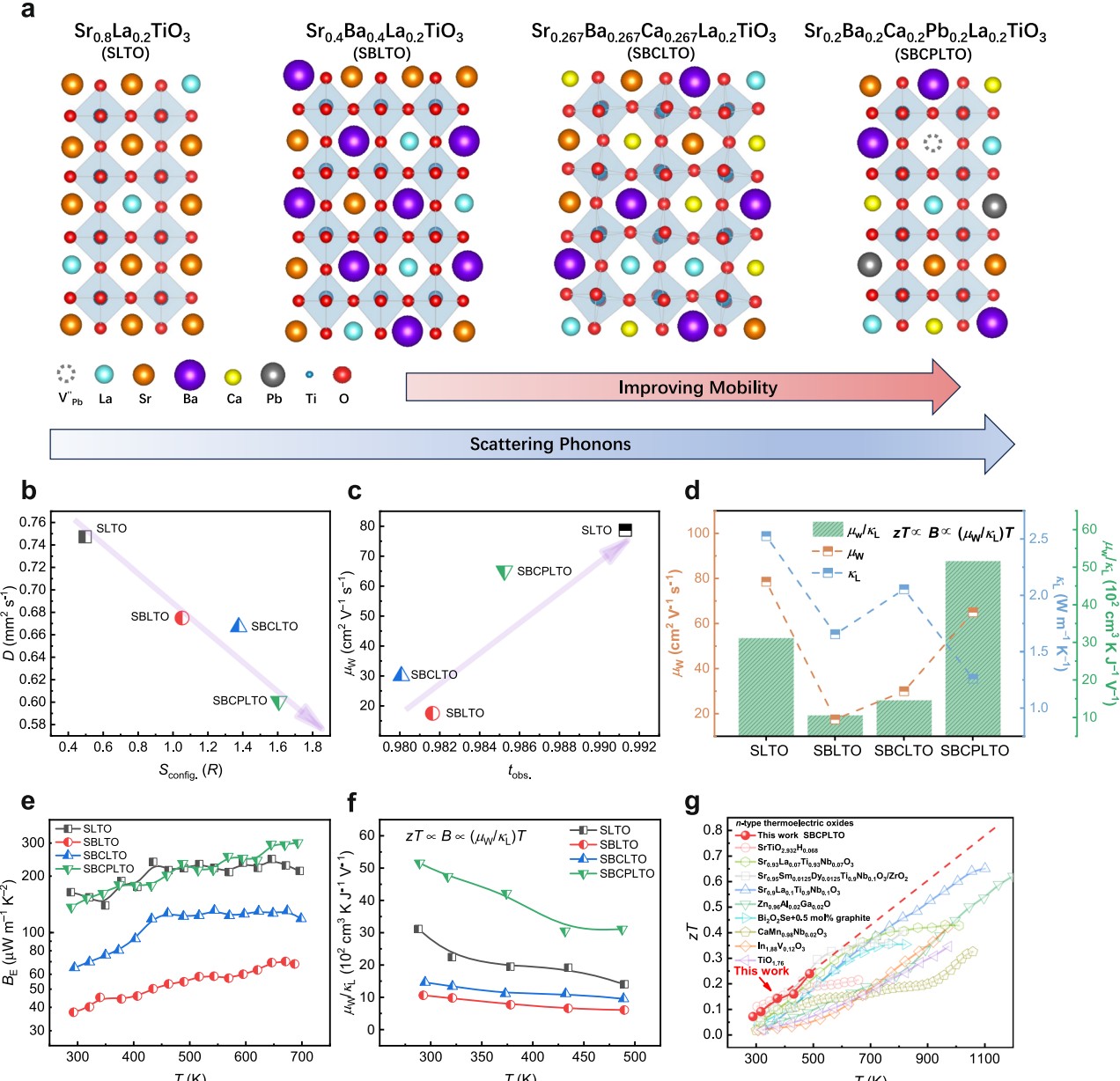

**Fig. 4 | Decoupled carrier-phonon transport and enhanced thermoelectric properties of entropy-engineered thin films. a** Schematic diagram of tuning A sites and $TiO_6$ octahedrons to decouple the carrier-phonon transport with increasing entropy. The symbols of elements were shown in the insets. **b** The correlation between the thermal diffusivity ($D$) and the nominal configuration entropy ($S_{config.}$). The thermal diffusivity was measured on corresponding unannealed bulks at 923 K for reference. The purple arrow is a guide for the eyes. **c** The relation between weighted mobility and observed tolerance factor $t_{obs.}$ to explain the structural origin of mobility recovery. The tolerance factor was calculated from the bond length measured by PDF on corresponding bulks for reference. The

purple arrow is a guide for the eyes. **d** The lattice thermal conductivity ($\kappa_L$), weighted mobility ($\mu_W$), and $\mu_W/\kappa_L$ of SLTO, SBLTO, SBCLTO, and SBCPLTO to show the extent of decoupling and effects of different elements when engineering entropy at room temperature. **e, f.** Temperature-dependent $B_E$ (in log scale) (**e**), $\mu_W/\kappa_L$ (**f**) of entropy-engineered thin films. **g** Temperature-dependent $zT$ of SBCPLTO, in comparison with other competitive $n$-type thermoelectric pure oxides[12,58–62,100–102]. The $zT$ measurement was limited by the highest allowed measuring temperature of TDTR. Since the PF and $\kappa$ are plateau-like with increasing $T$ at high temperatures, the $zT$ was extrapolated to high temperature (dashed red line).

entropy engineering in thermoelectrics and materials facing trade-offs among strongly coupled physical variables.

## Methods
### Sample preparation
The corresponding ceramic targets for film fabrication and the bulks for referencing characterization were fabricated by conventional solid-state reaction and pressureless sintering in air. The raw powers including $La_2O_3$ powder (99.99%, Aladdin, China), $TiO_2$ powder (99.8%, anatase, Aladdin, China), $SrCO_3$ powder (99.95%, Macklin, China), $BaCO_3$ powder

(99.95%, Macklin, China), $CaCO_3$ powder (99.99%, Macklin, China), $PbTiO_3$ powder (99.5%, Alfa Aesar, China), were stoichiometrically mixed ($PbTiO_3$ 5% excess) by ball milling with ethanol as agent. The dried mix powders were calcined for 3 h in muffle furnaces in air (1523 K for $(Sr_{0.8}La_{0.2})TiO_3$, $(Sr_{0.4}Ba_{0.4}La_{0.2})TiO_3$, $(Sr_{0.267}Ba_{0.267}Ca_{0.267}La_{0.2})TiO_3$, and 1323 K for $(Sr_{0.2}Ba_{0.2}Ca_{0.2}Pb_{0.2}La_{0.2})TiO_3$. The calcined powers were ball milled and were pressed into pellets (30 mm in diameter for targets). After cold isotropic pressing at 220 MPa, the pellets were sintered for 10 h in muffle furnaces in the air (1723 K for $(Sr_{0.8}La_{0.2})TiO_3$, $(Sr_{0.4}Ba_{0.4}La_{0.2})TiO_3$, $(Sr_{0.27}Ba_{0.27}Ca_{0.27}La_{0.2})TiO_3$, and 1523 K for

$(Sr_{0.2}Ba_{0.2}Ca_{0.2}Pb_{0.2}La_{0.2})TiO_3$, and the dense ceramic targets were in single phases (Supplementary Fig. 25). The corresponding bulk ceramic samples for reference were prepared at the same time with targets. The corresponding bulks were not annealed to remain electrically insulated to exclude the carrier contribution to thermal transport for reference.

The thin films were grown with nominal contents $(Sr_{0.8}La_{0.2})TiO_3$, $(Sr_{0.4}Ba_{0.4}La_{0.2})TiO_3$, $(Sr_{0.27}Ba_{0.27}Ca_{0.27}La_{0.2})TiO_3$, $(Sr_{0.2}Ba_{0.2}Ca_{0.2}Pb_{0.2}La_{0.2})TiO_3$ via pulsed laser deposition (PLD). The single crystal LSAT (001) substrates were used ($12 \times 4 \times 0.5$ mm$^3$, Hefei Kejing Materials Technology Co., Ltd., China), and the high-temperature silver paste (05001-AB, 0.5 oz, Spi Supplies, the U.S.) was applied to ensure uniform temperature when depositing films. The PLD was performed with a KrF laser (wavelength = 248 nm), and the energy density, repetition rate, number of pulses, growth temperature, working oxygen partial pressure were 0.8 J cm$^{-2}$, 5 Hz, 6000 pulses, 1023 K, 20 Pa, respectively. After deposition, the films were annealed in situ in oxygen (~600 Pa) for 20 min to improve the crystal quality and were cooled down to room temperature at a rate of 10 K min$^{-1}$. The as-grown thin films underwent annealing in 5% H$_2$/95% Ar mixed gas at 1173 K for 2 h to be reduced. After the annealing in reducing gas, the thin films turned into conductors, while the LSAT substrates which are not sensitive to oxygen vacancies, were tested to remain insulated, excluding the electrical contribution from the substrates[40].

## Measurement of transport behaviors

**Electrical transport.** The temperature-dependent electrical conductivity and Seebeck measurement of films were performed on the commercial apparatus MRS-3 (JouleYacht, China). The temperature-dependent carrier concentration ($n_H$) and the Hall mobility ($\mu_H$) were calculated from $\sigma_H$ ($\mu_H = \sigma_H|R_H|$) and $R_H$ ($n_H = 1/eR_H$) by van der Pauw method in Hall measurement (8408, Lake Shore Cryotronics, Inc.). The effective mass $m^*$ could be derived by combining Hall data with the Seebeck coefficient. In this work, the mobility of samples varies from 0.1 to 2.2 cm$^2$ V$^{-1}$ s$^{-1}$, right at the test limit of DC field measurement ($1–1 \times 10^6$ cm$^2$ V$^{-1}$ s$^{-1}$). Therefore, the AC field measurement developed by the Lake Shore Cryotronics[63,64], using AC magnetic field rather than the traditional DC mode, was applied to remove the effect of misalignment, to enlarge the measuring range of Hall mobility ($1 \times 10^{-3}$ to $1 \times 10^6$ cm$^2$ V$^{-1}$ s$^{-1}$) and to make the measurement more precise. According to the manual given by the Lake Shore official website, all field values in Hall measurement were nominal and could vary ±1%", and the error of $\mu_H$ could be calculated by the sum of the error (1%) of five field-related variants ($V_H$, $I$, $B$, $I'$, $U'$) to be 5%. The $\mu_H$–$T$ correlation with an error of 5% was plotted in Supplementary Fig. 27a. However, multiple factors could enlarge the error of Hall measurement, like measuring geometry, electrical contact, shape and size of metal pads, etc., and the error could even reach 20%. $\mu_H$–$T$ correlation with exaggerated errors of 20%, and 30% were plotted in Supplementary Figs. 27b and 27c. As can be seen in Supplementary Fig. 27, the $\mu_H$ differences between samples are still more significant than the error bar.

**Thermal transport.** The thermal properties of annealed LSAT(001) and annealed films were measured by the time-domain thermoreflectance (TDTR) technique[65,66]. TDTR utilizes a pump-probe configuration, where the pump beam heats the sample, and the probe beam monitors the transient sample surface temperature changes via thermoreflectance. The thermal properties are extracted by fitting the thermoreflectance signal with a thermal diffusion model. In this work, the pump beam was modulated with a frequency of 10.7 MHz, and focused on the sample surface together with the probe beam through a 10× objective lens, with a spot radius of 5.2 μm. Prior to the TDTR measurements, a 151 nm aluminum (Al) film was deposited on the film. The Al transducer layer thickness was determined by X-ray reflectance (XRR), and the thickness of the perovskite oxide film was determined

by cross-sectional SEM images. The surface of Al film could be oxidized with a native oxide layer (Al$_2$O$_3$) of 2–3 nm. The native oxide layer is commonly counted as an extra Al thickness of ~3 nm in other groups[67,68]. The 3 nm was added to the measured Al thickness (154 nm in total in this work), and the total thickness was used in the fitting process of the TDTR method. During the temperature-dependent thermal conductivity TDTR measurement, the chamber was vacuumed, and it could be assumed that the native oxide layer would not be further oxidized during high-temperature measurement. The approximation of adding 3 nm to the Al layer still worked in the measurement. The heat capacities of the Al transducer and the annealed LSAT (001) substrate were taken from literature value and laser flash analysis (LFA), respectively, and the heat capacities of perovskite oxide films were estimated by DSC and LFA. The thermal conductivity of the annealed LSAT substrate was measured separately using LFA ($\kappa = C_p\rho D$). It is important to clarify that the electrical transport in this work was measured in plane, while the thermal conductivity measured by the TDTR method was out of plane due to difficulty in technique. Additionally, the temperature range of the TDTR measurement was up to 473 K limited by the apparatus. Fortunately, for films whose thicknesses were ~ 200 nm, the mean free paths of phonons in entropy-engineered oxides were smaller by orders (in 10$^0$ Å), leading to little influence from transport confinement by thickness[25,69]. And for cubic or pseudo-cubic phases, the 2nd-order tensor-like thermal conductivity could be isotropic in the strain-released epitaxial oxides with few grain boundaries[70], the $zT$ could be roughly calculated despite measuring in different directions. Since the PF and the thermal conductivity displayed plateau-like behaviors with increasing temperature, the $zT$ value could be estimated by fitting the data linearly to high temperature. The laser flash apparatus (LFA, LFA457, NETZSCH, Germany) measured the thermal diffusivity and the thermal capacity of substrates and corresponding ceramics for reference, and standard samples were measured simultaneously to ensure validity. The transverse and longitudinal phonon group velocity of unannealed corresponding oxide ceramics were measured by ultrasonic pulse-echo method (Olympus 5072PR, Japan).

## Characterizations

**HAADF-STEM and EDS and diffraction.** The cross-sectional scanning transmission electron microscope (STEM) samples were acquired by focus ion beam (FIB) milling. The samples were thinned to 50 nm by FIB with a current of 240–50 pA under an accelerating voltage of 30 kV, and with 20 pA and 5 kV during polishing process. High angle annular dark field (HAADF), annular bright field (ABF), and energy dispersion spectroscopy (EDS) images along <110> zone axis with a high resolution of 0.059 nm were collected by a 200 kV JEOL ARM 200CF equipped with double aberration correctors. The convergence angle and the collection angle were set at 25 mrad, 48–2000 mrad, respectively. The exposure time when collecting EDS signals was 2 μs, and the total time was 90 min. The noise in ABF images was filtered by HREM-Filters released by HREM Research Inc.

**X-ray total scattering experiment.** X-ray diffraction measurements were made at beamline 3W1 of the Beijing Synchrotron Radiation (Beijing, China) using an incident X-ray beam of wavelength 0.206468 Å (60.05 keV). A Mercu 1717HS detector (2048 × 2048 pixels of a 140 ×140-μm CsI scintillator) was placed ~150 mm downstream of the sample, giving a $Q_{max}$ of 25 Å$^{-1}$. The setup was calibrated using the diffraction pattern from polycrystalline CeO$_2$ powder. The unannealed ceramic bulks $(Sr_{0.8}La_{0.2})TiO_3$, $(Sr_{0.4}Ba_{0.4}La_{0.2})TiO_3$, $(Sr_{0.27}Ba_{0.27}Ca_{0.27}La_{0.2})TiO_3$, $(Sr_{0.2}Ba_{0.2}Ca_{0.2}Pb_{0.2}La_{0.2})TiO_3$ were glued with Compton tapes to an aluminum alloy frame. The measurement procedure was controlled by iDetector software and 20 s exposure time was set for $(Sr_{0.4}Ba_{0.4}La_{0.2})TiO_3$, $(Sr_{0.27}Ba_{0.27}Ca_{0.27}La_{0.2})TiO_3$, and $(Sr_{0.2}Ba_{0.2}Ca_{0.2}Pb_{0.2}La_{0.2})TiO_3$, 10 s for $(Sr_{0.8}La_{0.2})TiO_3$ due to its

stronger signal. Background patterns were collected with the same setup and exposure time. The raw diffraction data were reduced from two-dimensional images and corrected for the effects of polarization and geometry using the program Fit2D[71] and absorption, geometry, detector effects, and the normalization procedure carried out using PDFgetX2[72].

**Reverse Monte Carlo (RMC) simulation.** Using RMC simulation, the structural models were derived from the diffraction data[73]. To start, the configuration of ~5000 atoms was used. Constraints were set, including the connectivity of Ti-O bonds (i. e. Ti was coordinated to a reasonable number of O up to 2.5 Å) and the atom-atom approaches. The aim of setting the constraints was to avoid the results of unrealistic structures in physics. By counting the atomic configurations generated by RMC simulations, the structural information was derived.

**SHG.** Second harmonic generation (SHG) spectroscopy was performed under a home-designed SHG microscope[74]. The excitation laser was provided by a Ti: sapphire mode-locking femtosecond laser (MaiTai SP, Spectra-Physics) which generated 35 fs pulses with a repetition rate of 80 MHz and a wavelength centered at 800 nm. The fundamental laser beam was directed onto the sample at normal incidence and focused to a focal spot diameter of ≈1 μm using a 50× (numerical aperture NA = 0.55) objective. The back-scattered SHG signal was collected with the same objective and detected using a Hamamatsu photomultiplier tube. The SHG microscopy images of the square areas selected were acquired by scanning the probing position using a pair of galvanometers. When scanning the interested region, the angle between the polarization direction of the incident light and the SHG light collected was fixed the same at 90°. The rate of scanning was ~50 ms pixel$^{-1}$ and ~2 μm s$^{-1}$.

**Other regular characterizations.** The out-of-plane X-ray diffraction (XRD) in two theta-omega mode and the reciprocal space mapping (RSM) of perovskite films were conducted by X-ray diffractor (PANalytical X'Pert MRD, Netherlands) to characterize the epitaxy nature and crystal quality. The cross-sectional scanning electron microscopy (SEM, MERLIN VP Compact, ZEISS, Germany) was used to obtain the thickness of films. Cross-sectional morphology and microstructure were observed by transmission electron microscopy (TEM, JEM-2100F, JEOL, Japan) to analyze the epitaxy nature, lattice fringes, dislocation analysis by FFT, and the energy dispersive spectra (EDS) of films. The focused ion beam (FIB, TESCAN S9000X, Brno, Czech) was applied to thin the samples. Geometric phase analysis[75–77] (GPA, HREM Research Inc.) plug-in helped the strain analysis and mapping from HRTEM images of films. Raman spectroscopy (HORIBA, Japan) was utilized to get information on bond vibrations and crystal symmetry, and the laser wavelength was 325 nm for thin films to minimize the signals of substrates and 532 nm for bulks. The information of bond length and charge valence were recorded by X-ray photoelectron spectroscopy (XPS, Thermo Fisher ESCALAB 250Xi, the U.S.), and the bandgaps were measured by ultraviolet-visible light spectrophotometer (UV-vis, L950, PerkinElmer, the U.S.) absorption mode on films. The rough concentration of elements in films, especially Pb, was derived from the Electron probe X-ray micro analyzer (EPMA, JXA8230, JEOL, Japan) by the Pb: Ti ratio.

## Calculation
**Band structure DFT.** The electron density of states (DOS), crystal orbital Hamilton population (COHP)[78], and band structures were calculated through density functional theory (DFT) based first-principle scheme using the Vienna ab-initio Simulation Package (VASP)[79–81]. The electron-ion interaction was modeled by projector-augmented wave (PAW) method[82], and the Perdew–Burke–Ernzerhof (PBE) generalized gradient approximation (GGA)[83] was used to simulate the electron interactions. The doping elements of La, Ca, Ba, and Pb are treated by using virtual crystal approximation (VCA). The cut-off energy was 400 eV for all the computations. The O and Pb vacancies were not considered in the calculation.

**Phonon dispersion calculation.** First, we trained a deep learning potential model to describe the potential energy surface between atoms. The dataset was generated by DP-GEN[84,85], using $3 \times 3 \times 3$ and $2 \times 2 \times 2$ supercells of the cubic primitive cells, and the $2 \times 1 \times 1$ supercell of the tetragonal conventional cell of $SrTiO_3$ as approximation[86,87] (Supplementary Fig. 28 and CIF files in Supplementary Data 1–4), and labeled through first-principles calculations, consisting of a total of 2028 data points. The method and supercell size have been applied in phonon dispersion calculation of $SrTiO_3$ in other works[86,87]. Furthermore, in the self-consistent phonon calculations by Tadano et al. in 2015, a $2 \times 2 \times 2$ supercell is sufficient to obtain phonon dispersion close to the experimental results[88]. The first-principle calculations were performed using the VASP package[79,80,89,90] with the projector-augmented plane wave (PAW)[82] method and the PBEsol functional[83,91]. The energy cutoff was set to 700 eV, and the K-point grid spacing was set to 0.14 Å$^{-1}$. Next, utilizing the labeled dataset, we employed DeePMD-kit[92] to train a neural network. Compared to the results obtained from DFT calculations, the energy accuracy achieved $1.21 \times 10^{-3}$ eV per atom. Using the deep learning potential model, we applied the SCAILD[93,94] method to calculate the interatomic force constants of $SrTiO_3$ at 300 K within a $5 \times 5 \times 5$ supercell. To introduce mass disorder, structures were randomly generated for $SrTiO_3$, $(Sr_{0.8}La_{0.2})TiO_3$, $(Sr_{0.4}Ba_{0.4}La_{0.2})TiO_3$, $(Sr_{0.27}Ba_{0.27}Ca_{0.27}La_{0.2})TiO_3$, $(Sr_{0.2}Ba_{0.2}Ca_{0.2}Pb_{0.2}La_{0.2})TiO_3$ subsequently basing on the supercell (CIF files in Supplementary Data 5–9). Finally, the UPHO[95,96] package was then employed to compute the unfolded phonon spectrum by combining the IFCs and the doped structure with the lowest energy for each composition. In SCAILD and UPHO calculation, the api of phonopy package was used[97,98]. The O and Pb vacancies were not considered in the calculation.

### Reporting summary
Further information on research design is available in the Nature Portfolio Reporting Summary linked to this article.

## Data availability
The authors declare that the data supporting the findings of this study are available within the paper and its Supplementary Information files. The source data used in this study are available in the Figshare database under accession code https://doi.org/10.6084/m9.figshare.26797660. Any other relevant data are also available upon reasonable request from Y.-H. L.

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

## Acknowledgements

This work was financially supported by the Basic Science Center Project of the National Natural Science Foundation of China under grant No. 52388201 (Y.-H. Lin), and the National Science Foundation of China under grant No. 52172211 (M.Z.). This work (C.S.) was also supported by the National Key Basic Research Program of China (grant No. 2020YFA0406101). Q.Z. and L.G. were supported by the NSFC (grant No. 52025025 and No. 52072400). Z.Z. acknowledges financial support from the Shuimu Tsinghua Scholar Program and the China Postdoctoral Science Foundation under grant No. 2024T170479. W.X. was supported by the National Natural Science Foundation of China (grant No. 12075273). J.-L.L. acknowledges the National Natural Science Foundation of China under grant No. 52372075. B.W. acknowledges the support of the National Science Foundation of China (grant No. 12304040), China Postdoctoral Science Foundation (grant No. 2023M742004), and the State Key Laboratory of New Ceramic and Fine Processing Tsinghua University (grant No. KF202304). PDF measurements were conducted at beamline 3W1 of the Beijing Synchrotron Radiation (Beijing, China). We also acknowledge the contribution of 4B9B beamline of Beijing Synchrotron Radiation (Beijing, China) for the exploration on XAS measurement. We thank the Beijing PARATERA Tech CO., Ltd. (https://cloud.paratera.com). We thank Kaiqi Nie, Zunqiu Xiao, Xiaomin Jia, Jiasheng Guo, Haojie Han, and Shun Lan for fruitful discussions.

## Author contributions

Y.-H. Lin and Y. Zheng conceived the study. Y. Zheng, Z. Zhou, and Y.-H. Lin wrote the manuscript. Y. Zheng conducted the experiments and analyzed the data in this study. Q. Zhang and L. Gu performed the Cs-corrected STEM. C. Shi, W. Xu, Y. Zhang, and D. Yang carried out the PDF measurement, and C. Shi analyzed the results by reverse Monte Carlo simulation. Y. Lu, W. Ma, and X. Zhang measured thermal conductivity by TDTR. J. Han and C. Lin calculated the phonon dispersion. H. Chen and D. Yi helped with the calculation of energy bands. Y. Ma and Q. Li supported by SHG mapping. Y. Yang, B. Wei, B. Yang, M. Zou, W. Zhang, C. Liu, L. Dou, J.-L. Lan, and C.-W. Nan assisted in the writing of the manuscript. All authors discussed and revised the manuscript.

## Competing interests

The authors declare no competing interests.
