## [Peer Review File · Nature Communications]

Carrier-phonon decoupling in perovskite thermoelectrics via entropy engineeringEditorial Note: Parts of this Peer Review File have been redacted as indicated to remove third-party material where no permission to publish could be obtained.

REVIEWER COMMENTS

Reviewer #1 (Remarks to the Author):

The manuscript entitled "Carrier-phonon decoupling in perovskite thermoelectrics via entropy engineering", authored by Yunpeng Zheng et al., realized carrier-phonon decoupling in perovskite thermoelectrics, and achieved a remarkable zT value among n-type oxide thermoelectrics, providing effective solutions to improve carrier mobility and effective mass via entropy design in thermoelectrics with low thermal conductivity.

The previous works in the field of high-entropy thermoelectrics mainly focus on suppressing thermal conductivity and improving the density of state effective mass, while this work obtained increasing carrier mobility with increasing entropy and decreasing thermal conductivity. The authors added local disorder to crystals, and simultaneously tuned the ion displacement and structural symmetry, thus enabling the greatly enhanced $\mu W/\kappa L$ of $\sim 5.2 \times 10^3 \text{ cm}^3 \text{ K J}^{-1} \text{ V}^{-1}$ and the estimated zT of ~ 0.8 at 1173 K in the high-entropy (Sr_{0.2}Ba_{0.2}Ca_{0.2}Pb_{0.2}La_{0.2})TiO₃ film, outperforming the other n-type thermoelectric oxides. The work points out new direction on decoupling carrier-phonon transport to improve zT, and inspires the design of thermoelectrics.

The manuscript is well written and easy to read, which the reviewer appreciates. Considering the novelty and the improvement of performances, the manuscript deserves positive consideration for Nature Communications after minor revisions. There are still several points to be further explained which are listed as follows:

1. The authors selected A-sites of perovskites to design entropy engineered thermoelectrics, the reason for the crystal-site selection should be added. What about B-site entropy engineering?
2. The authors fabricated SrTiO₃-based thin films on LSAT by PLD methods, and measured the thermal conductivity by TDTR, which were both complicated, the reason why the authors chose to prepare films (not ceramics) should be added.
3. The O vacancies and Pb vacancies in the films made the analysis more difficult, what is the role of these two kinds of defects?
4. The lattice thermal conductivity of SBCLTO (with higher entropy than SBLTO) was higher than SBLTO, how to explain?
5. The valence of Ti, the electronegativity of A-site dopants (influencing the electron distribution), and the tilting of BO₆ octahedrons might contribute to the electrical transport. Why could the authors attribute the mobility difference to Ti displacement?

Reviewer #2 (Remarks to the Author):

This paper reports the decoupling of electron-phonon transport and ZT enhancement in SrTiO₃ films by high entropy approach. Mechanism of the electron-phonon transport decoupling was discussed based on various analysis methods and DFT calculations, reaching to the reasonable conclusion. Unfortunately, the ZT of the high-entropy SrTiO₃ is not high (~ 0.24 at 488 K), but this paper should provide a new insight for the design of high performance thermoelectric oxides. So I think this paper can be possibly considered for the publication in Nature Commun. In order to improve the paper quality, I suggest a number of

questions and comments:

1. In abstract, authors showed only the estimated ZT values to exaggerate their results. The actual (measured) ZT values should be shown as well.

2. Page. 5 in introduction part: Decoupling of electron-phonon transport has been reported in SrTiO₃. Authors should cite these works. Examples are listed as follows,

Y. Lin et al., ACS Appl. Mater. Interfaces 7 (2015) 15898–908.

X. Feng et al., Carbon 112 (2017) 169–176.

M. T. Dylla et al., Adv. Mater. Interfaces 6 (2019) 1900222.

O. Okhay et al., Carbon 143 (2019) 215–222.

J. U. Rahman et al., Sci. Rep. 9 (2019) 8624.

J. Cao et al., ACS Appl. Mater. Interfaces 13 (2021) 11879-11890.

X. He et al., Adv. Funct. Mater. 33 (2023) 2213144.

3. Why thermal conductivity of SLTO films was very low at 2.52 W/(mK), compared to pure STO with 11W/mK? If this value is true, phonon scattering by other than high entropy effect seems to be more effective in reducing lattice thermal conductivity. In addition, authors should show TDTR spectra in Supporting Information.

4. Fig. 1B: Why lattice thermal conductivity increases with increasing temperature. Oxidation of Al transducer layer does not affect the measurement by TDTR?

4. Authors synthesized high-entropy oxide bulk polycrystals, as shown in Fig.S25. Why authors did not measure electronic and thermal transport properties of the bulks? If possible, authors should verify that thermal conductivity can be reduced also in the bulks.

5. Fig. 2B: Hall voltage was measured correctly at high temperature? Error bars would be necessary.

6. Fig. 4G: Authors should select the ZT references for state-of-the-art oxides, correctly. Some oxides exhibit ZT over 0.1 even at room temperature.

Y. Lin et al, ACS Appl. Mater. Interfaces 7 (2015) 15898–15908.

A. Kumar et al., J. Materiomics 9 (2023) 191–196.

X. He et al., Adv. Funct. Mater. 33 (2023) 2213144.

Higher ZT at high temperature

M. Ito et al., Scripta Mater. 48 (2003) 403–408.

C.-O. Romo-De-La-Cruz et al., Chem. Mater. 32 (2020) 9730–9739.

H. Bakhshi et al., Dalton Trans. 49 (2020) 17–22

J.-B. Li et al., J. Mater. Chem. C 6 (2018) 7594–7603.

Reviewer #3 (Remarks to the Author):

Report: Carrier-Phonon Decoupling in Perovskite Thermoelectrics via Entropy Engineering"

The main subject of the manuscript is the thermoelectric properties, which are the ability of a material to transform energy into heat and vice versa.

The goal of the authors is to contribute to the advancement of new thermoelectric materials, and they propose an entropy engineering strategy to realize carrier-phonon decoupling in the

SrTiO₃ (STO) perovskite, and show that this methodology may be a promising approach to achieve the "figure of merit" (zT) in thermoelectric materials.

1. SrTiO₃ is a well-known material that has been studied extensively by experimentalists and theoreticians, so it is essential to discuss the findings in the existing literature on SrTiO₃. Below are listed, among others, some relevant articles that should be included in the bibliography and perhaps include the second reference as a comparison effect:

i) <https://doi.org/10.1016/j.ceramint.2022.01.262>

ii) <https://doi.org/10.1016/j.commat.2023.112274>

2. The simulations' lack of information can make reproducing the results difficult. Perhaps additional information can be added to the supplementary material.

i) Was a supercell model used in the computational model? How big is it? Please add some figures.

ii) If the response to the questions above is YES, are there some convergence tests for a supercell model? What are the convergence parameters for the calculation?

iii) Where is the CIF file for each adopted model?

iv) Was the Phonopy software used for the phonon calculation? See the cited reference.

v) The total density of states does not bring relevant pieces of information. I suggest COHP analysis: <https://pubs.acs.org/doi/10.1021/jp202489s>

vi) I understand that collaborative experiments/theoretical projects are both important branches of research. Nevertheless, both approaches should be discussed at the same high level.

3. Supplementary Material:

Figure S21 and S22: Please do not make the E-E_f set at 0 eV (minimum of the conduction band). It is recommended that the Fermi level not be set to 0 eV. Several valuable pieces of information can be extracted from this analysis.

Figure S23c It is recommended that the Fermi level not be set to 0 eV. What is the atom of site A (SLTO, SBLTO, SBCLTO)?

Figure S23f Please add more data with zT values greater than the zT value of SrTiO₃ and include zT values obtained for other doping studies.

Reviewer #4 (Remarks to the Author):

Response to Reviewers

Reviewer #1 (Remarks to the Author):

The manuscript entitled “Carrier-phonon decoupling in perovskite thermoelectrics via entropy engineering”, authored by Yunpeng Zheng *et al.*, realized carrier-phonon decoupling in perovskite thermoelectrics, and achieved a remarkable zT value among n -type oxide thermoelectrics, providing effective solutions to improve carrier mobility and effective mass *via* entropy design in thermoelectrics with low thermal conductivity.

Response:

Thanks very much for the reviewer's recognition on our work “**providing effective solutions**” to realize “carrier-phonon decoupling” and achieve “a remarkable zT value among n -type oxide thermoelectrics”.

The previous works in the field of high-entropy thermoelectrics mainly focus on suppressing thermal conductivity and improving the density of state effective mass, while this work obtained increasing carrier mobility with increasing entropy and decreasing thermal conductivity. The authors added local disorder to crystals, and simultaneously tuned the ion displacement and structural symmetry, thus enabling the greatly enhanced μ_W/κ_L of $\sim 5.2 \times 10^3 \text{ cm}^3 \text{ K J}^{-1} \text{ V}^{-1}$ and the estimated zT of ~ 0.8 at 1173 K in the high-entropy $(\text{Sr}_{0.2}\text{Ba}_{0.2}\text{Ca}_{0.2}\text{Pb}_{0.2}\text{La}_{0.2})\text{TiO}_3$ film, outperforming the other n -type thermoelectric oxides. The work points out a new direction on decoupling carrier-phonon transport to improve zT , and inspires the design of thermoelectrics.

Response:

Thanks very much for the reviewer's recognition. As the reviewer stated, the new point of the work is to increase carrier mobility with increasing entropy, “**pointing out a new direction on decoupling carrier-phonon transport to improve zT , and inspires the**

design of thermoelectrics”.

The manuscript is well written and easy to read, which the reviewer appreciates. Considering the novelty and the improvement of performances, **the manuscript deserves positive consideration for *Nature Communications* after minor revisions.**

There are still several points to be further explained which are listed as follows:

Response:

Thanks very much for the reviewer's **recommendation of publication**. The authors have responded to the questions raised point by point as follows, and have enriched the manuscript in the paper.

1. The authors selected A-sites of perovskites to design entropy engineered thermoelectrics, the reason for the crystal-site selection should be added. What about B-site entropy engineering?

Response:

Thanks very much for the reviewer's question. The reason for the selection of A sites, rather than B sites, to engineer entropy, is stated below. Since the Ti 3d orbitals form the conduction band minimum (CBM) deciding the electron transport in *n*-type SrTiO₃-based semiconductors¹, and A-site vibrations correspond to acoustic phonon branches and low frequency optical branches dominating lattice thermal conductivity^{2,3}, the A-site entropy engineering could help maintain the TiO₆ octahedrons undisturbed to prevent mobility deterioration and meanwhile strongly scatter phonons⁴⁻⁶, realizing carrier-phonon decoupling.

When multiple elements are introduced at B sites, despite the reduction in thermal conductivity, the electrical performances might be harmed. In one of the unpublished

works, the authors have fabricated B-site high-entropy ceramics $\text{Sr}(\text{Ti}_{0.2}\text{Zr}_{0.2}\text{Hf}_{0.2}\text{Sn}_{0.2}\text{Nb}_{0.2})\text{O}_3$ and $\text{Sr}(\text{Ti}_{0.2}\text{Zr}_{0.2}\text{Hf}_{0.2}\text{Sn}_{0.2}\text{Ta}_{0.2})\text{O}_3$, and found that though the lattice thermal conductivity was suppressed to $\sim 1.9 \text{ W m}^{-1} \text{ K}^{-1}$, the electrical conductivity deteriorated to $10^{-2} \text{ S cm}^{-1}$ at $\sim 973 \text{ K}$ even after strong electrical current assisted reducing process⁷. Other works on B-site high-entropy perovskite oxides are consistent with our results, with suppressed thermal conductivity and decreased electrical conductivity^{8,9}. The disorder and distortion of BO_6 octahedrons and the smaller amount of Ti might contribute to the deterioration of electrical properties.

In conclusion, A-site entropy design was the key point of carrier-phonon decoupling in this work, and the site selection is important when engineering entropy in high-entropy ceramics. The discussion on site selection has been added to the manuscript.

2. The authors fabricated SrTiO_3 -based thin films on LSAT by PLD methods, and measured the thermal conductivity by TDTR, which were both complicated, the reason why the authors chose to prepare films (not ceramics) should be added.

Response:

Thanks very much for the reviewer's question. The reason for using the thin films, rather than bulks, is that the high-quality thin films grown by PLD method could reduce the influence of grain boundaries in perovskites on electrical and thermal transport, making it easier to construct entropy-transport relationship and enhance the thermoelectric properties.

It is difficult to rule out grain boundary effects in bulks. In fact, the authors synthesized high-entropy oxide bulk polycrystals as PLD targets and to provide thermal parameters for reference. The efforts on the electrical transport measurement of high-entropy bulks were made in our previous work⁷. In that work, due to the negative-charge A-site

vacancies (especially Pb vacancies) at grain boundaries forming potential barrier for electrons, which is very common in donor-doped SrTiO₃ polycrystals¹⁰, the electrical transport was dominated by grain boundary scattering and the electrical conductivity deteriorated (**Fig. R1**)¹¹. Additionally, the sluggish diffusion effect of high-entropy ceramics could hinder the uniform annealing and the densification of bulks, making it difficult to improve the electrical property of the polycrystal bulks. And on the other hand, the phonon grain boundary scattering is also significant, making it complex to clarify the effect of entropy on thermal conductivity. To clarify the entropy engineering effect and to enhance the electrical performances, the authors chose to fabricate epitaxial thin films and optimize the quality to rule out extrinsic effects¹².

Compared to bulks, the extrinsic effects are less significant in epitaxial thin films. When it comes to the substrate contribution on electrical conductivity, after the annealing in reducing gas, the thin films turned into conductors, while the LSAT substrates which are not sensitive to oxygen vacancies, were tested to remain insulated, excluding the electrical contribution from the substrates¹³. And in this work, the thermal conductivity of the corresponding bulks was also measured by LFA for reference (**Fig. S24** and **Fig. R2**), the trend of thermal conductivity change in films and bulks were consistent, and the values were close. In summary, the decision to use films to explore entropy effects is appropriate, and the relevant discussion has been added in the manuscript.

[Redacted]

Fig. R1 | Carrier grain boundary scattering effect of *n*-type SrTiO₃-based polycrystalline ceramics by Y. Lin *et al.*¹¹

Fig. R2 | The lattice thermal conductivity of SLTO, SBLTO, SBCLTO, SBCPLTO and corresponding unannealed bulks at 473 K for comparison. The minimum thermal conductivity of SBCPLTO was plotted for reference.

3. The O vacancies and Pb vacancies in the films made the analysis more difficult, what is the role of these two kinds of defects?

Response:

Thanks very much for the reviewer's question. Both oxygen vacancies and Pb vacancies could play important role in our entropy engineered perovskite thin films.

O vacancies: from the aspect of electrical transport, the formation of oxygen vacancies could provide electrons to Ti $3d$ orbitals which consist of the conduction band minimum, leading to increase of the carrier concentration and electrical conductivity^{12,14}, and meanwhile, the oxygen vacancies could also tune the size and the distortion of TiO_6 octahedrons, thus influencing the mobility. As for thermal transport, the increasing amount of oxygen vacancies, as a part of entropy increase, could increase the rate of phonon scattering, further suppressing thermal conductivity. Limited by the characterization, the exact amount of oxygen vacancies was hard to measure, but the reducing process among samples with different entropies was kept the same to try to create similar amount of oxygen vacancies.

Pb vacancies: from the aspect of electrical transport, the formation of Pb vacancies could decrease the electron concentration, but at the same time, the smaller Pb vacancies could further tune the A-site size towards SrTiO_3 and tolerance factor towards 1, from SBCLTO to SBCPLTO, thus suppressing Ti displacement and enhance the carrier mobility. Simultaneously, the Pb could provide $6p$ orbitals at CBM to increase the DOS, but the volatilization of Pb could weaken the effect. Also, the Pb vacancies could introduce mass and size fluctuation, strongly scattering phonons to suppress the thermal conductivity.

The effect of vacancies could also be considered as part of entropy effect, comprehensively tuning the transport behaviors, and the discussion on vacancies has

been added in the manuscript.

4. The lattice thermal conductivity of SBCLTO (with higher entropy than SBLTO) was higher than SBLTO, how to explain?

Response:

Thanks very much for the reviewer's question. In this work, the thermal conductivity of the corresponding bulks was also measured by LFA for reference (**Fig. S24 and Fig. R2**). The lattice thermal conductivity κ_L (SLTO) > κ_L (SBCLTO) > κ_L (SBLTO) > κ_L (SBCPLTO) were both found in films and the corresponding bulks, and results were close between LFA and TDTR, bulks and films, indicating that the measurement is reliable. To explain, though the entropy of SBCLTO is higher than SBLTO, it is worth noting that the thermal capacity was not necessarily monotonically correlated with entropy, and the Ca with smaller mass could lead to larger thermal capacity (Dulong-Petit Law and Neumann-Kopp Rule¹⁵), thus larger thermal conductivity in SBCLTO than that in SBLTO. The related discussion and information were added in the revised supporting information.

5. The valence of Ti, the electronegativity of A-site dopants (influencing the electron distribution), and the tilting of BO₆ octahedrons might contribute to the electrical transport. Why could the authors attribute the mobility difference to Ti displacement?

Response:

Thanks very much for the reviewer's question. The Ti displacement is the main factor affecting the carrier mobility.

For the valence of Ti, the valence of Ti remained as +4 in samples with different entropies and before and after the reducing process, and thus the valence of Ti could

not be the main factor influencing the mobility. To verify, in Ti 2*p* XPS spectra (**Fig. 2D**), there were no obvious Ti³⁺ shoulder peaks near Ti⁴⁺ in all films¹⁶, suggesting that the Ti mainly existed as Ti⁴⁺, which was also ensured by XANES Ti *L*₃ edge spectrum sensitive to valence^{17,18} in **Fig. R3**.

For the electronegativity of A-site dopants, the electronegativity could affect the electron density distribution, and with the higher electronegativity of A-site elements, electrons are more likely to distribute around A-site, rather than Ti sites, thus reducing the connectivity of electron cloud between Ti-O network. However, the electronegativity $\eta(\text{Pb}) > \eta(\text{Ca}) > \eta(\text{Sr}) > \eta(\text{Ba})$, while the mobility $\mu_w(\text{Pb}) \approx \mu_w(\text{Sr}) > \mu_w(\text{Ca}) > \mu_w(\text{Ba})$, which is not consistent with the trend of electronegativity and the scenario. The electronegativity seems not to be the main factor.

As for the tilting of TiO₆ octahedrons, according to **Table S10** and **Fig. S19**, the TiO₆ octahedrons in SLTO are tilted, and in SBCPLTO, the tilting is ignorable. However, the weighted mobility is similar between SLTO and SBCPLTO. Furthermore, the TiO₆ tilting is ignorable in SBLTO, but the mobility deteriorates significantly, which means that the tilting is not the main factor affecting carrier mobility.

As for Ti displacement, the displacement of Ti ions decreased from SBLTO to SBCLTO, and Ti ions hardly displaced in SBCPLTO (**Fig. 3D-F**). The Ti displacement dominated the electron transport in our entropy engineered perovskite titanates, and with increasing displacement of Ti, the weighted mobility decreased (**Fig. 3G**). To explain, the displaced ions could create shorter and longer bonds, and the electrons could be localized in shorter bonds with overlapping electron clouds¹⁹ (**Fig. 3A**). From the aspect of orbitals, the breaking of the symmetry by Ti displacement could result in irregular bond lengths and O-Ti-O angles departing from 180°, and thus the split of *t*_{2g} orbitals would lead to carrier localization^{20,21,22}. Furthermore, the inhomogeneous displacement

among adjacent octahedrons in disordered medium-entropy samples could contribute to inhomogeneous electronic potentials, thus scattering electrons. After improving the symmetry, the similar bond feature could enable homogeneous electron cloud overlapping among Ti-O bonds, delocalizing the electron to transport. In summary, the Ti displacement was the distortion that hindered the electron transport in this case. The relevant discussion on the factors affecting mobility has been added to the manuscript.

Fig. R3 | The synchrotron XANES Ti L₃ edge spectrum of SBCPLTO.

Reviewer #2 (Remarks to the Author):

This paper reports the decoupling of electron-phonon transport and *ZT* enhancement in SrTiO₃ films by high entropy approach. Mechanism of the electron-phonon transport decoupling was discussed based on various analysis methods and DFT calculations, reaching to the reasonable conclusion.

Response:

Thanks very much for the reviewer's recognition on our work "decoupling electron-

phonon transport” and achieving “ ZT enhancement in SrTiO_3 films by high entropy approach”. **“Reasonable conclusion” on “mechanism of the electron-phonon transport decoupling”** was made according to various methods.

Unfortunately, the ZT of the high-entropy SrTiO_3 is not high (~ 0.24 at 488 K), but this paper should **provide a new insight for the design of high performance thermoelectric oxides**. So I think this paper can be **possibly considered for the publication in Nature Communications**. In order to improve the paper quality, I suggest a number of questions and comments:

Response:

Thanks very much for the reviewer’s recognition on our work **“providing a new insight for the design of high performance thermoelectric oxides”** and the possible **consideration for the publication in Nature Communications**. Though the zT value is lower than state-of-the-art thermoelectric alloys, the advancement of $zT \sim 0.24$ of SrTiO_3 at 488 K is competitive among n -type thermoelectric oxides (intrinsic performance without compositing). Considering that SrTiO_3 is a kind of classic model materials, the findings on mechanisms of decoupling carrier-phonon transport could provide insight for high-entropy thermoelectrics. To improve the quality of this paper, the authors respond to the comments point by point, and add more discussions to enrich the manuscript in the revised version with highlights.

1. In abstract, authors showed only the estimated ZT values to exaggerate their results. The actual (measured) ZT values should be shown as well.

Response:

Thanks very much for the reviewer’s suggestion. The SrTiO_3 -based samples could at

least endure the temperature of 1173 K (annealing temperature~1173 K, and the bulks were calcinated and sintered at 1523 K and 1723 K). Limited by the temperature range where the apparatus could reach, especially limited by the TDTR measurement, the measured zT could only be acquired at ~488 K, far lower than the temperature SrTiO₃ could endure. Since the PF and the thermal conductivity displayed plateau-like behaviors with increasing temperature, the zT value could be estimated by fitting the data linearly to high temperature. For sure, it is more professional to show the measured zT in the abstract with the estimated one, and the authors appreciate the reviewer's suggestion to make the paper solid. The authors have added the measured zT value in the abstract in the revised version of manuscript.

2. Page. 5 in introduction part: Decoupling of electron-phonon transport has been reported in SrTiO₃. Authors should cite these works. Examples are listed as follows,

Y. Lin et al., ACS Appl. Mater. Interfaces 7 (2015) 15898–908.

X. Feng et al., Carbon 112 (2017) 169–176.

M. T. Dylla et al., Adv. Mater. Interfaces 6 (2019) 1900222.

O. Okhay et al., Carbon 143 (2019) 215–222.

J. U. Rahman et al., Sci. Rep. 9 (2019) 8624.

J. Cao et al., ACS Appl. Mater. Interfaces 13 (2021) 11879-11890.

X. He et al., Adv. Funct. Mater. 33 (2023) 2213144.

Response:

Thanks very much for the reviewer's important complements. Interface engineering using carbon-based materials could help transport carriers and block phonons both in alloys²³ and in oxides²⁴, thus decoupling carrier-phonon transport. The first six references recommended by the reviewer modified the grain boundaries of SrTiO₃-

based ceramics with carbon-based modifier, synergistically tuning the electrical-thermal transport from the extrinsic aspect (compositing)²⁵⁻³⁰, which should be cited and has been added in the manuscript by authors. It is not quite common to decouple carrier-phonon transport in SrTiO₃ by intrinsically tuning the composition and structure. For example, in H introduced SrTiO₃ reported by X. He *et al.*³¹, the μ_W and κ_L decreased simultaneously. In comparison, the entropy engineered SrTiO₃-based thin films displayed improved mobility with entropy and minimized thermal conductivity, providing new insights from the aspect of intrinsic modulation. The discussion on the previous reports on SrTiO₃ has been enriched in the revised version.

3. Why thermal conductivity of SLTO films was very low at 2.52 W/(mK), compared to pure STO with 11 W/mK? If this value is true, phonon scattering by other than high entropy effect seems to be more effective in reducing lattice thermal conductivity. In addition, authors should show TDTR spectra in Supporting Information.

Response:

Thanks very much for the reviewer's question. To verify, in Fig. S24 (i. e. Fig. R4), the lattice thermal conductivity of corresponding bulks was measured by LFA, and the results between TDTR and LFA, bulks and films, were close to some extent. The acoustic and low-frequency optical branches of phonons are mainly influenced by A sites in perovskites², and thus the doping at A sites with different mass, radius and charge and the vacancies could effectively scatter phonons^{32,33}. For example, J. Wang *et al.* fabricated La, Nb co-doped Sr_{0.9}La_{0.1}Ti_{0.9}Nb_{0.1}O₃ ceramics¹⁴, and the lattice thermal conductivity was suppressed significantly from 11 W m⁻¹ K⁻¹ to ~3.8 W m⁻¹ K⁻¹. In this work, high concentration (20%) of La³⁺ was introduced to A sites of SrTiO₃-based perovskites. The high doping level, the higher valence of +3, the large mass, and

the small size of La^{3+} could bring about large fluctuation and rich phonon scattering sources. Additionally, since the A sites were nominally fully occupied, and the La (+3), Sr(+2), Ba(+2), Ca(+2), Pb(+2) have no other smaller valence states, when performing pulsed laser deposition, the A-site vacancies tended to form. The introduction of A-site vacancies and thus large charge and mass disorder in La-doped SrTiO_3 -based ceramics could also cause strong phonon scattering^{3,25,34}. Since 3.3% occupation of vacancies at A sites in $\text{Sr}_{0.9}\text{La}_{0.067}\text{TiO}_3$ could strongly scatter phonons²⁵, the amount of A-site vacancies coming from donor doping and low-pressure deposition could effectively scatter phonons as well. Also, the interfaces of thin films could also act as phonon scatters at low temperature and at room temperature²⁵. On the other hand, the decrease of thermal conductivity could be significant with slight fluctuation when the thermal conductivity is high. For instance, in BN with high thermal conductivity, the existence of isotopes could suppress thermal conductivity.

After the thermal conductivity is primarily reduced, it is difficult but important to continue to suppress the thermal conductivity by some strategies. Further increasing entropy at A sites could almost halve the lattice thermal conductivity from $\sim 2.52 \text{ W m}^{-1} \text{ K}^{-1}$ to $\sim 1.25 \text{ W m}^{-1} \text{ K}^{-1}$, which proves the effectiveness of high-entropy strategy. From STO to SLTO to SBCPLTO, the entropy increases with mass and size disorder, and stabilized defects, contributing to the phonon scattering intrinsically and extrinsically. Additionally, the TDTR spectra at room temperature are shown in revised supporting materials (Fig. S26, Fig. R5).

Fig. R4 | The lattice thermal conductivity of SLTO, SBLTO, SBCLTO, SBCPLTO and corresponding unannealed bulks at 473 K for comparison. The minimum thermal conductivity of SBCPLTO was plotted for reference.

Fig. R5 | The TDTR experimental data (10.7 MHz) and fitting curves. **A.** TDTR of SLTO, SBLTO, SBCLTO and SBCPLTO at 300 K. **B.** TDTR of SBCPLTO at different temperatures.

4. Fig. 1B: Why lattice thermal conductivity increases with increasing temperature.
 Oxidation of Al transducer layer does not affect the measurement by TDTR?

Response:

Thanks very much for the reviewer's question. The phenomenon of increasing lattice thermal conductivity with increasing temperature could be found easily in high-entropy materials and in amorphous phases³⁵, which is a sign of phonon scattering within adjacent neighboring atoms. Additionally, when the charge, mass and size disorder are strong enough, the glass-like phonon-glass thermal transport behaviors could be observed. For example, in $\text{Mg}_{0.167}\text{Ni}_{0.167}\text{Cu}_{0.167}\text{Co}_{0.167}\text{Zn}_{0.167}\text{Cr}_{0.167}\text{O}$, the charge disorder is responsible for the strong reduction in thermal conductivity, for the lattice distorted to compensate for the local charge imbalance³⁵. In low-entropy $\text{La}_{0.5}\text{K}_{0.5}\text{TiO}_3$ ³⁴ and $\text{La}_{0.5}\text{Na}_{0.5}\text{TiO}_3$ ³, despite the low entropy of $\sim 0.69R$, the 50% occupation of La and K/Na, the large valence difference (La^{3+} compared to K^+ and Na^+), size disorder (the radius of La^{3+} (1.36 Å) compared to the radius of K^+ (1.64 Å) and Na^+ (1.39 Å) with coordination number of XII), and mass disorder (La^{3+} (139) compared to the radius of K^+ (39) and Na^+ (23)) lead to phonon-glass behaviors. The introduction of A-site vacancies and thus large charge and mass disorder in La-doped SrTiO_3 -based ceramics could also cause strong phonon scattering^{3,25,34}. In this work, high concentration (20%) of La^{3+} was introduced to A sites of SrTiO_3 -based perovskites. The high doping level, the higher valence of +3, the large mass, and the small size of La^{3+} could bring about large fluctuation and rich phonon scattering sources. Additionally, since the A sites were nominally fully occupied, and the La (+3), Sr(+2), Ba(+2), Ca(+2), Pb(+2) have no other smaller valence states, when performing pulsed laser deposition, the A-site vacancies tended to form. Since 3.3% occupation of vacancies at A sites in $\text{Sr}_{0.9}\text{La}_{0.067}\text{TiO}_3$ could induce the phenomenon of increasing lattice thermal conductivity with increasing temperature²⁵, the amount of A-site vacancies coming from donor doping and low-pressure deposition could effectively

scatter phonons to approach the amorphous transport state as well. Also, the interfaces of thin films could also act as phonon scatters at low temperature and at room temperature²⁵. In summary, the glass-like phonon transport behaviors could be found in perovskites with strong A-site charge, mass and size disorder by high level of doping. The charge disorder functioning by the charge compensation and lattice distortion, could also be considered as part of size disorder. The discussion on the charge disorder and the glass-like phonon transport have been added in the revised manuscript. The authors are sorry that the quantitative characterization of defects in thin films is difficult due to the substrates providing strong signals, and there is still huge room for the thin film thermal measurement to be more accurate. The clearer explanation could be given through future deeper exploration in the field of entropy engineered thermoelectrics.

In terms of the detail of measurement and TDTR fitting, when measuring thermal conductivity by TDTR, a thin layer of aluminum (Al) was deposited by thermal evaporation on each sample to serve as a transducer. Al film thicknesses are obtained from the X-ray reflectance (XRR) to be ~151 nm. The surface of Al film could be oxidized with native oxide layer (Al_2O_3) of 2-3 nanometers. The native oxide layer is commonly be counted as extra Al thickness of ~3nm in other groups^{36,37}. The 3 nm was added to the measured Al thickness (154 nm in total in this work), and the total thickness was used in the fitting process of TDTR method. During the temperature-dependent thermal conductivity TDTR measurement, the chamber was vacuumed, and it could be assumed that the native oxide layer would not be further oxidized during high-temperature measurement. The approximation of adding 3 nm to Al layer still worked in the measurement. The discussion related to the TDTR measurement has been added in the revised version of the manuscript.

5. Authors synthesized high-entropy oxide bulk polycrystals, as shown in Fig.S25. Why authors did not measure electronic and thermal transport properties of the bulks? If possible, authors should verify that thermal conductivity can be reduced also in the bulks.

Response:

Thanks very much for the reviewer's question. The authors synthesized high-entropy oxide bulk polycrystals as PLD targets and to provide thermal parameters for reference. The efforts on the electrical transport measurement of high-entropy bulks were made in our previous work⁷. In that work, due to the negative-charge A-site vacancies (especially Pb vacancies) at grain boundaries forming potential barrier for electrons, which is very common in donor-doped SrTiO₃ polycrystals¹⁰, the electrical transport was dominated by grain boundary scattering and the electrical conductivity deteriorated (**Fig. R1**). Additionally, the sluggish diffusion effect of high-entropy ceramics could hinder the uniform annealing and the densification of bulks, making it difficult to improve the electrical property of the polycrystal bulks. To clarify the entropy engineering effect and to enhance the electrical performances, the authors chose to fabricate epitaxial thin films and optimize the quality to rule out extrinsic effects¹². The thermal conductivity was also measure in the previous work⁴. And in this work, the thermal conductivity of the corresponding bulks was also measured by LFA for reference (Fig. S24 and **Fig. R4**), the trend of thermal conductivity change in films and bulks were consistent, and the values were close. For example, The lattice thermal conductivity κ_L (SLTO) > κ_L (SBCLTO) > κ_L (SBLTO) > κ_L (SBCPLTO) were both found in films and the corresponding bulks (Fig. S24), and results were close between LFA and TDTR, bulks and films. (It is worth noting that the thermal capacity was not necessarily monotonically correlated with entropy, and the Ca with smaller mass could

lead to larger thermal capacity, thus larger thermal conductivity in SBCLTO than that in SBLTO). According to the measurement, the thermal conductivity could be reduced both in films and in bulks. The related discussion and information were added in the revised supporting information.

6. Fig. 2B: Hall voltage was measured correctly at high temperature? Error bars would be necessary.

Response:

Thanks very much for the reviewer's suggestion. It has long been a difficult point to measure the Hall voltage of thermoelectric oxides with high carrier concentration ($\sim 10^{21} \text{ cm}^{-3}$) and low carrier mobility ($\sim 10^{-1} - 10^0 \text{ cm}^2 \text{ V}^{-1} \text{ s}^{-1}$), for the Hall voltage and Hall coefficient ($V_H = IB/ned$, $R_H = 1/ne$) in this case could be too small to detect, and any disturbance during measurement could significantly affect the results of Hall voltage³⁸. In this work, the mobility of samples varies from 0.1-2.2 $\text{cm}^2 \text{ V}^{-1} \text{ s}^{-1}$, right at the test limit of DC field measurement (1 to $1 \times 10^6 \text{ cm}^2 \text{ V}^{-1} \text{ s}^{-1}$). Therefore, the authors decided to use the AC field measurement developed by the Lake Shore Cryotronics^{39,40} (Related information: <https://www.lakeshore.com/products/categories/specification/material-characterization-products/hall-effect-systems/8400-series-hms>), using AC magnetic field rather than the traditional DC mode, to remove the effect of misalignment, to enlarge the measure range of Hall mobility (1×10^{-3} to $1 \times 10^6 \text{ cm}^2 \text{ V}^{-1} \text{ s}^{-1}$) and to make the measurement more precise. After the application of AC field measurement, the Hall measurement could be performed and reasonable data could be obtained. However, the disturbance at high temperature is more obvious, and the acoustic phonon scattering dominated mobility decreases with increasing temperature, thus making it difficult to measure the Hall voltage at high temperature. The authors had to abandon some high-

temperature data points severely deviating from the trend after checking that the Hall voltage was not accurately measured, which explains the absence of some data points at high temperature.

In terms of error bar, according to the manual given by Lake Shore official website, “all field values are nominal and can vary $\pm 1\%$ ”, and $\mu_H = V_H d \sigma / IB$, $\sigma = I / U' ad$, the error of μ_H could be calculated by the sum of the error (1%) of five field-related variants (V_H , I , B , I' , U') to be 5%. The $\mu_H - T$ correlation with error of 5% was plotted in **Fig. R6A**. As can be seen in **Fig. R6A**, the 5% error bar is not notable. However, multiple factors could enlarge the error of Hall measurement, like measuring geometry, electrical contact, shape and size of metal pads, etc., and the error could even reach 20%. The authors also plotted $\mu_H - T$ correlation with error of 20%, and 30% to exaggerate in **Fig. R6B** and **Fig. R6C**. As can be seen in **Fig. R6**, the μ_H differences between samples are still larger than the error bar, and the relative value relationship of μ_H , $\mu_H(\text{SBCPLTO}) \approx \mu_H(\text{SLTO}) > \mu_H(\text{SBCLTO}) > \mu_H(\text{SBLTO})$, still holds, consistent with the weighted mobility μ_w calculated from electrical conductivity and Seebeck coefficient ($\mu_w(\text{SBCPLTO}) \approx \mu_w(\text{SLTO}) > \mu_w(\text{SBCLTO}) > \mu_w(\text{SBLTO})$) (**Fig. 2C**). In summary, the mobility change by tuning Ti displacement is more significant than the error. Since the value of the error is difficult to define, and the error bars could make the figure hard to identify, the **Fig. R6** has been added in supporting information and the discussion on the accuracy of Hall measurement was added in the manuscript and method part.

Fig. R6 | The temperature variant Hall mobility with error bar. A. 5% (calculated error) error bar. B. and C. Exaggerated error bar with error 20% (B.) and 30% (C.).

7. Fig. 4G: Authors should select the ZT references for state-of-the-art oxides, correctly.

Some oxides exhibit ZT over 0.1 even at room temperature.

Y. Lin et al, ACS Appl. Mater. Interfaces 7 (2015) 15898–15908.

A. Kumar et al., J. Materiomics 9 (2023) 191–196.

X. He et al., Adv. Funct. Mater. 33 (2023) 2213144.

Higher ZT at high temperature

M. Ito et al., Scripta Mater. 48 (2003) 403–408.

C.-O. Romo-De-La-Cruz et al., Chem. Mater. 32 (2020) 9730–9739.

H. Bakhshi et al., Dalton Trans. 49 (2020) 17–22.

J.-B. Li et al., J. Mater. Chem. C 6 (2018) 7594–7603.

Response:

Thanks very much for the reviewer's suggestion. The authors have added several references of state-of-the-art *n*-type thermoelectric oxides in the **Fig. 4G (Fig. R7)**. The authors need to explain that our work mainly focuses on how to improve the thermoelectric performances of *n*-type pure oxides from the intrinsic aspect of tuning the composition and crystal structure *via* entropy engineering. The field this work in is defined as *n*-type thermoelectric oxides, and thus *p*-type oxides or carbon-based composites, in authors' opinions, are on quite different tracks, though they are promising directions to apply and verify the effect of entropy engineering to achieve higher *zT*. For the works exhibit *zT* over 0.1 even at room temperature, Y. Lin *et al.* displayed the thermoelectric performance of SrTiO₃-graphene composites²⁵. Though the high-temperature *zT* of the composites is not attractive, the high *zT* at room temperature mainly originated extrinsically from graphene of so large amount (0.6 wt%, not mol%), not SrTiO₃ intrinsically. Especially the lattice thermal conductivity of less than 1 (less than amorphous limit) increased with temperature (In the work done by K. Koumoto *et al.*, even when the grain size was suppressed to 50 nm, the lattice thermal conductivity decreased with increasing temperature⁴¹), showing that graphene dominated the transport in these carbon-based composites. Therefore, the authors decided not to cite this work focusing on compositing effects. A. Kumar *et al.* also did great job on room-temperature high-entropy *p*-type LaCoO₃-based thermoelectric oxides⁴², but is not in the scope of *n*-type thermoelectric oxides. X. He *et al.* really did impressive work on H-doped SrTiO₃-based thin films with excellent room-temperature

zT ⁴³, and the authors are sorry to forget to cite this reference and added the work in the **Fig. 4G (Fig. R7)**. And for the work with good zT values at high temperature, the works done by M. Ito *et al.*⁴⁴ and C.-O. Romo-De-La-Cruz *et al.*⁴⁵ were on p -type oxides $\text{Na}_x\text{Co}_2\text{O}_4$ and $\text{Ca}_3\text{Co}_4\text{O}_9$, also out of the scope of n -type thermoelectric oxides. The works of H. Bakhshi *et al.*⁴⁶ and J.-B. Li *et al.*⁴⁷ are both on SrTiO_3 -based ceramics of high zT at high temperature, for which the authors are sorry to miss. The references of SrTiO_3 -based thermoelectric oxides of high zT at high temperature and of different doping levels have been added in **Fig. 4G (Fig. R7)** in the revised version of manuscript^{48,49,50,51,52,53,43,47,46}. As can be seen in **Fig. R7**, SBCPLTO is among the best-performance n -type thermoelectric oxides at the same temperatures.

Fig. R7 | Temperature-dependent zT of SBCPLTO, in comparison with other competitive n -type thermoelectric oxides^{48,49,50,51,52,53,43,47,46}.

Reviewer #3 (Remarks to the Author):

The main subject of the manuscript is the thermoelectric properties, which are the ability of a material to transform energy into heat and vice versa.

The goal of the authors is to contribute to the advancement of new thermoelectric materials, and they propose an entropy engineering strategy to realize carrier-phonon decoupling in the SrTiO₃ (STO) perovskite, and show that this methodology may be a **promising approach** to achieve the "figure of merit" (zT) in thermoelectric materials.

Response:

Thanks very much for the reviewer's recognition on our work contributing to "advancement of new thermoelectric materials", realizing "carrier-phonon decoupling", and proposing "promising approach to achieve high zT in thermoelectric materials". Entropy engineering could simultaneously suppress thermal conductivity and tune the electrical transport, which is effective to enhance thermoelectric figure of merit zT via carrier-phonon decoupling.

1. SrTiO₃ is a well-known material that has been studied extensively by experimentalists and theoreticians, so it is essential to discuss the findings in the existing literature on SrTiO₃. Below are listed, among others, some relevant articles that should be included in the bibliography and perhaps include the second reference as a comparison effect:

i) <https://doi.org/10.1016/j.ceramint.2022.01.262>

ii) <https://doi.org/10.1016/j.commat.2023.112274>

Response:

Thanks very much for the reviewer's kind advice, which could improve the quality of the manuscript significantly. SrTiO₃ is a classic research material in the fields of

dielectrics, photonics, and thermoelectrics for its ideal cubic structure and high tunability. Extensively studied by experimentalists and theoreticians, the existing literatures on SrTiO₃ need to be comprehensively reviewed. The authors have reviewed part of articles relevant to thermoelectric SrTiO₃, but reminded by the reviewer, the literature review was limited to experimental works, and the discussion needs to be enriched. The relevant references which the reviewer recommended studied the thermoelectric performances of SrTiO₃ from computational aspects. By first-principle calculation, the first reference stated that the multi-elements doping might improve electrical-thermal transport synergistically, supporting our findings, and the second reference proposed that in some cases, individual dopant Ag could influence the energy band structure, like the function of Pb in SBCPLTO in our work. The authors have added the references the reviewer recommended, and enriched the discussion on SrTiO₃ in the introduction part and main text from both experimental and computational aspects. Please see the highlights in the revised manuscript.

2. The simulations' lack of information can make reproducing the results difficult.

Perhaps additional information can be added to the supplementary material.

i) Was a supercell model used in the computational model? How big is it? Please add some figures.

ii) If the response to the questions above is YES, are there some convergence tests for a supercell model? What are the convergence parameters for the calculation?

iii) Where is the CIF file for each adopted model?

iv) Was the Phonopy software used for the phonon calculation? See the cited reference.

v) The total density of states does not bring relevant pieces of information. I suggest

COHP analysis: 10.1021/jp202489s

vi) I understand that collaborative experiments/theoretical projects are both important branches of research. Nevertheless, both approaches should be discussed at the same high level.

Response:

Thanks very much for the reviewer's kind advice, additional information has been added in the part of experimental methods. The point-by-point response to the questions are listed as follows:

i) When training the deep learning potential model, DP-GEN was used to generate a dataset^{54,55}, using $3\times 3\times 3$ and $2\times 2\times 2$ supercells of the cubic primitive cells, and the $2\times 1\times 1$ supercell of the tetragonal conventional cell of SrTiO_3 as approximation^{56,57}. After the training, the structures were randomly generated basing on the supercell. When calculating the phonon dispersion, the doped $5\times 5\times 5$ structure with lowest energy for each composition was selected to conduct the calculation. The figures of the supercells have been added in supporting materials (**Fig. S28** and **Fig. R8**). The CIF files of the supercell have also been attached as supporting materials. (file name: $2\times 2\times 2$ _cubic_supercell_POSCAR_STO_221, $3\times 3\times 3$ _cubic_supercell_POSCAR_STO_221, $2\times 1\times 1$ _tetra_supercell_POSCAR_140, $5\times 5\times 5$ _cubic_POSCAR)

Fig. R8 | SrTiO₃ supercell used for deep learning potential model as approximation. A. cubic 2×2×2; B. cubic 3×3×3; C. tetragonal 3×3×3; D. 5×5×5.

ii) When generating the dataset for the deep learning potential model, limited by the computing power, the authors did not try to use a larger supercell. However, this method and supercell size have been applied in phonon dispersion calculation of SrTiO₃ in other works^{56,57}. Furthermore, in the self-consistent phonon calculations by Tadano *et al.* in 2015, a 2×2×2 supercell is sufficient to obtain phonon dispersion close to the experimental results⁵⁸.

iii) The CIF of the adopted model have been attached as parts of supporting information (file name: STO_POSCARD, SLTO_POSCARD, SBLTO_POSCARD, SBCLTO_POSCARD, SBCPLTO_POSCARD).

iv) In SCAILD and UPHO calculation, the authors did use the api of Phonopy package. The authors should have cited relevant reference^{59,60}, and the reference was added in

the revised version.

v) Thanks for the reviewer's kind suggestion. COHP calculation⁶¹ has been conducted, and has been added in the supporting information to enrich the paper. From the COHP result in **Fig. R9A (Fig. S29A)**, the negative part of COHP below Fermi level represents bonding states, and the positive part over Fermi level represents antibonding states. Like the results got from DOS calculation, the antibonding and bonding states near the Fermi level mainly contribute to the Ti-O antibonding and bonding in SLTO, SBLTO, SBCLTO, and the A-side d orbitals, Ti *d* orbitals and Pb 6*p* orbitals participates in the states near the Fermi level in SBCPLTO. It could be speculated that the density of states of SBCPLTO is enhanced from the higher COHP at Fermi level. And since the $i\text{COHP}$ $\text{SLTO} < \text{SBLTO} < \text{SBCLTO} < \text{SBCPLTO} < 0$ at Fermi level (**Fig. R9A**), the composition is thermodynamically stable, and the bonds of SBCPLTO is estimated to be more soft than other bonds. Limited by the VCA method, the COHP could only be taken as reference for trends, which the authors are sorry for. Related discussion has been added in the manuscript and in the supporting information.

Fig. R9 | COHP calculation. A. COHP calculation; **B.** integrated $i\text{COHP}$ calculation. The Fermi level is marked by dotted line in the same color with curves of corresponding composition.

vi) Thanks for the reviewer's kind advice. The content of theoretical projects has been added including details, discussion and figures. The calculation helps explain the increase of phonon scattering and the increase of density of states. The topic of this manuscript is on the carrier-phonon decoupling, so the authors have included related theoretical figures in the manuscript, and the part on DOS was arranged in the supporting information.

3. Supplementary Material:

Figure S21 and S22: Please do not make the $E-E_F$ set at 0 eV (minimum of the conduction band). It is recommended that the Fermi level not be set to 0 eV. Several valuable pieces of information can be extracted from this analysis.

Response:

Thanks very much for the reviewer's suggestion. The statement of " $E-E_F$ set at 0 eV" in the manuscript is an unclear expression by authors, and the authors are so sorry for the misunderstandings caused by the unclearness. $E-E_F$ equals 0 when carrier energy is Fermi level, and the Y-axis denoting $E-E_F$ is widely accepted by researchers^{62,63}. Due to the donor doping by La at A sites, the calculated Fermi level lies right above the conduction band minimum (not exactly the CBM), which means that there was no manual setting on the energy. However, just as the reviewer recommended, if the Y-axis could be originally plotted as E , not $E-E_F$, several valuable pieces of information can be extracted from this analysis. The authors have revised the figure and change the axis from $E-E_F$ to E , and many thanks to pointing out that. According to the calculated energy dispersion band and DOS in **Fig. S21-S23 (Fig. R10 and Fig. R11)**, the Fermi level is in the conduction band, meaning that the donor doping effectively provides electrons in conduction bands. From the position of Fermi level, information of doping level, effective mass, mobility and degeneracy could be given.

Fig. R10 | The energy band dispersion calculation results. A. SLTO. B. SBLTO. C. SBCLTO. D. SBCPLTO. The Fermi level is marked by dotted line.

Fig. R11 | The DOS of entropy engineered perovskites. A. Total DOS. The main orbital contribution is marked in the figure. B. Partial density of states (DOS) of entropy engineered thin films. Only A-site d , Ti $3d$, and Pb $6p$ orbitals were plotted for simplification. The atom of site A is Sr, La for SLTO; Sr, Ba, La for SBLTO; Sr, Ba, Ca, La for SBCLTO; Sr, Ba, Ca, Pb, La for SBCPLTO. The Fermi level is marked by dotted line in the same color with curves of corresponding composition.

Figure S23C It is recommended that the Fermi level not be set to 0 eV. What is the atom of site A (SLTO, SBLTO, SBCLTO)?

Response:

Thanks very much for the reviewer's suggestion. Like the response above, sorry for the unclear expression, and the Y-axis has been adapted to E without subtracting E_F with more information provided (**Fig. S23C**, **Fig. R11**). The atom of site A is Sr, La for SLTO; Sr, Ba, La for SBLTO; Sr, Ba, Ca, La for SBCLTO; Sr, Ba, Ca, Pb, La for SBCPLTO. The definition of A has been added to the figure legend of **Fig. S23C**. Thanks for the reviewer's detailed suggestion.

Figure S23F Please add more data with zT values greater than the zT value of SrTiO_3 and include zT values obtained for other doping studies.

Response:

Thanks very much for the reviewer's suggestions, and sorry for the insufficient literature review. The data with zT values greater than this work on n -type thermoelectric oxides have been added, and SrTiO_3 with other doping levels have included in Fig. S23F^{43,46,47}. The adapted figure is shown below as **Fig. R12**, and the expression in the manuscript has also been adapted. The zT of entropy engineered SBCPLTO is still among the best at room temperature and higher temperatures in n -type thermoelectric oxides.

Fig. R12 | The zT_{ave} . Average zT_{ave} (RT~473 K) comparison among SBCPLTO and other competitive *n*-type thermoelectric oxides^{48,49,50,51,52,53,43,46,47}.

Reviewer #4 (Remarks to the Author):

Response:

Thanks very much for the reviewer's comments above. The authors have responded to the above comments point by point, and made major revisions to the manuscript.

Reference:

1. Dylla, M. T., Kang, S. D. & Snyder, G. J. Effect of Two-Dimensional Crystal Orbitals on Fermi Surfaces and Electron Transport in Three-Dimensional Perovskite Oxides. *Angew. Chemie - Int. Ed.* **58**, 5503–5512 (2019).
2. Popuri, S. R. *et al.* Phonon-Glass and Heterogeneous Electrical Transport in A-

- Site-Deficient SrTiO₃. *J. Phys. Chem. C* **123**, 5198–5208 (2019).
3. Daniels, L. M. *et al.* Phonon-glass electron-crystal behaviour by A site disorder in n-type thermoelectric oxides. *Energy Environ. Sci.* **10**, 1917–1922 (2017).
 4. Zheng, Y. *et al.* Electrical and thermal transport behaviours of high-entropy perovskite thermoelectric oxides. *J. Adv. Ceram.* **10**, 377–384 (2021).
 5. Zhang, P. *et al.* High-entropy MTiO₃ perovskite oxides with glass-like thermal conductivity for thermoelectric applications. *J. Alloys Compd.* **937**, 168366 (2023).
 6. Zhang, P. *et al.* Reduced lattice thermal conductivity of perovskite-type high-entropy (Ca_{0.25}Sr_{0.25}Ba_{0.25}RE_{0.25})TiO₃ ceramics by phonon engineering for thermoelectric applications. *J. Alloys Compd.* **898**, 162858 (2022).
 7. Zheng, Y. *et al.* Electrical and thermal transport behaviours of high-entropy perovskite thermoelectric oxides. *J. Adv. Ceram.* **10**, 377–384 (2021).
 8. Lou, Z. *et al.* A novel high-entropy perovskite ceramics Sr_{0.9}La_{0.1}(Zr_{0.25}Sn_{0.25}Ti_{0.25}Hf_{0.25})O₃ with low thermal conductivity and high Seebeck coefficient. *J. Eur. Ceram. Soc.* **42**, 3480–3488 (2022).
 9. Maiti, T. *et al.* High-entropy perovskites: An emergent class of oxide thermoelectrics with ultralow thermal conductivity. *ACS Sustain. Chem. Eng.* **8**, 17022–17032 (2020).
 10. Huang, J. *et al.* Simultaneously Breaking the Double Schottky Barrier and Phonon Transport in SrTiO₃-Based Thermoelectric Ceramics via Two-Step Reduction. *ACS Appl. Mater. Interfaces* **12**, 52721–52730 (2020).

11. Lin, Y. *et al.* Expression of interfacial Seebeck coefficient through grain boundary engineering with multi-layer graphene nanoplatelets. *Energy Environ. Sci.* **13**, 4114–4121 (2020).
12. Zheng, Y. *et al.* Electrical Property Enhancement in Orientation-Modulated Perovskite La-Doped SrTiO₃ Thermoelectric Thin Films. *Adv. Funct. Mater.* **33**, 1–11 (2023).
13. Scullin, M. L. *et al.* Anomalously large measured thermoelectric power factor in Sr_{1-x}La_xTiO₃ thin films due to SrTiO₃ substrate reduction. *Appl. Phys. Lett.* **92**, 1–4 (2008).
14. Wang, J. *et al.* Record high thermoelectric performance in bulk SrTiO₃ via nano-scale modulation doping. *Nano Energy* **35**, 387–395 (2017).
15. Leitner, J., Voňka, P., Sedmidubský, D. & Svoboda, P. Application of Neumann-Kopp rule for the estimation of heat capacity of mixed oxides. *Thermochim. Acta* **497**, 7–13 (2010).
16. Watts, F. & Wolstenholme, J. *Surface Analysis by XPS and AES. Book* vol. 1 (John Wiley & Sons, 2015).
17. Li, T. *et al.* Ultrahigh-Efficiency Superior Energy Storage in Lead-Free Films with a Simple Composition. *J. Am. Chem. Soc.* **146**, 1926–1934 (2024).
18. Sefat, A. S., Amow, G., Wu, M. Y., Botton, G. A. & Greedan, J. E. High-resolution EELS study of the vacancy-doped metal/insulator system, Nd_{1-x}TiO₃, x = 0 to 0.33. *J. Solid State Chem.* **178**, 1008–1016 (2005).
19. Jiang, B. *et al.* High figure-of-merit and power generation in high-entropy

- GeTe-based thermoelectrics. *Science (80-.)*. **377**, 208–213 (2022).
20. Tilley, R. J. D. *Perovskites: Structure-Property Relationships. Perovskites: Structure-Property Relationships* (John Wiley & Sons, 2016).
doi:10.1002/9781118935651.
 21. Bjaalie, L., Janotti, A., Himmetoglu, B. & Van De Walle, C. G. Turning SrTiO₃ into a Mott insulator. *Phys. Rev. B - Condens. Matter Mater. Phys.* **90**, 195117 (2014).
 22. Khomskii, D. I. *Transition Metal Compounds. Transition Metal Compounds* (2014). doi:10.1017/CBO9781139096782.
 23. Zhou, Z. *et al.* Compositing effects for high thermoelectric performance of Cu₂Se-based materials. *Nat. Commun.* **14**, 2410 (2023).
 24. Zhou, Z. *et al.* Boosting Thermoelectric Performance via Weakening Carrier-Phonon Coupling in BiCuSeO-Graphene Composites. *Small Methods* **2301619**, 1–7 (2024).
 25. Lin, Y. *et al.* Thermoelectric Power Generation from Lanthanum Strontium Titanium Oxide at Room Temperature through the Addition of Graphene. *ACS Appl. Mater. Interfaces* **7**, 15898–15908 (2015).
 26. Feng, X. *et al.* Graphene promoted oxygen vacancies in perovskite for enhanced thermoelectric properties. *Carbon N. Y.* **112**, 169–176 (2017).
 27. Dylla, M. T., Kuo, J. J., Witting, I. & Snyder, G. J. Grain Boundary Engineering Nanostructured SrTiO₃ for Thermoelectric Applications. *Adv. Mater. Interfaces* **6**, 1–7 (2019).

28. Okhay, O. *et al.* Thermoelectric performance of Nb-doped SrTiO₃ enhanced by reduced graphene oxide and Sr deficiency cooperation. *Carbon N. Y.* **143**, 215–222 (2019).
29. Rahman, J. U. *et al.* Grain Boundary Interfaces Controlled by Reduced Graphene Oxide in Nonstoichiometric SrTiO_{3-δ} Thermoelectrics. *Sci. Rep.* **9**, 1–12 (2019).
30. Cao, J. *et al.* Modulation of Charge Transport at Grain Boundaries in SrTiO₃: Toward a High Thermoelectric Power Factor at Room Temperature. *ACS Appl. Mater. Interfaces* **13**, 11879–11890 (2021).
31. He, X. *et al.* Hydride Anion Substitution Boosts Thermoelectric Performance of Polycrystalline SrTiO₃ via Simultaneous Realization of Reduced Thermal Conductivity and High Electronic Conductivity. *Adv. Funct. Mater.* **33**, (2023).
32. Braun, J. L. *et al.* Charge-Induced Disorder Controls the Thermal Conductivity of Entropy-Stabilized Oxides. *Adv. Mater.* **30**, 1–8 (2018).
33. Lim, M. *et al.* Influence of mass and charge disorder on the phonon thermal conductivity of entropy stabilized oxides determined by molecular dynamics simulations. *J. Appl. Phys.* **125**, (2019).
34. Daniels, L. M. *et al.* A and B site doping of a phonon-glass perovskite oxide thermoelectric. *J. Mater. Chem. A* **6**, 15640–15652 (2018).
35. Braun, J. L. *et al.* Charge-induced disorder controls the thermal conductivity of entropy-stabilized oxides. *Adv. Mater.* **30**, 1–8 (2018).
36. Park, K. & Bayram, C. Impact of dislocations on the thermal conductivity of

gallium nitride studied by time-domain thermoreflectance. *J. Appl. Phys.* **126**, (2019).

37. Zheng, Q. *et al.* Thermal conductivity of GaN, $\langle \text{GaN} \rangle$, and SiC from 150 K to 850 K. *Phys. Rev. Mater.* **3**, 014601 (2019).
38. Putley, E. H. & Rice, S. A. The Hall Effect and Related Phenomena. *Phys. Today* **15**, 72–72 (1962).
39. Lindermuth, D. J. R. An introduction to AC field Hall effect measurements. *J. Cryst. Growth* **36**, 29–35 (1976).
40. Lindemuth, J., Dodrill, B., Meyer, J. & Vurgaftman, I. Extraction of Low Mobility, Low Conductivity Carriers from Field. 48–50 (2002).
41. Koumoto, K., Wang, Y., Zhang, R., Kosuga, A. & Funahashi, R. Oxide Thermoelectric Materials: A Nanostructuring Approach. *Annu. Rev. Mater. Res.* **40**, 363–394 (2010).
42. Kumar, A., Drago, D., Berardan, D. & Drago, N. Thermoelectric properties of high-entropy rare-earth cobaltates. *J. Mater.* **9**, 191–196 (2023).
43. He, X. *et al.* Hydride Anion Substitution Boosts Thermoelectric Performance of Polycrystalline SrTiO₃ via Simultaneous Realization of Reduced Thermal Conductivity and High Electronic Conductivity. *Adv. Funct. Mater.* **33**, (2023).
44. Ito, M., Nagira, T., Furumoto, D., Katsuyama, S. & Nagai, H. Synthesis of Na_xCo₂O₄ thermoelectric oxides by the polymerized complex method. *Scr. Mater.* **48**, 403–408 (2003).

45. Romo-De-La-Cruz, C. O., Chen, Y., Liang, L., Williams, M. & Song, X. Thermoelectric oxide ceramics outperforming single crystals enabled by dopant segregations. *Chem. Mater.* **32**, 9730–9739 (2020).
46. Bakhshi, H., Sarraf-Mamoory, R., Yourdkhani, A., Abdelnabi, A. A. & Mozharivskyj, Y. Highly dense $\text{Sr}_{0.95}\text{Sm}_{0.0125}\text{Dy}_{0.0125}\text{Ti}_{0.025}\text{Ti}_{0.90}\text{Nb}_{0.10}\text{O}_{3\pm\delta}/\text{ZrO}_2$ composite preparation directly through spark plasma sintering and its thermoelectric properties. *Dalt. Trans.* **49**, 17–22 (2019).
47. Li, J. B. *et al.* Broadening the temperature range for high thermoelectric performance of bulk polycrystalline strontium titanate by controlling the electronic transport properties. *J. Mater. Chem. C* **6**, 7594–7603 (2018).
48. Wang, J. *et al.* Record high thermoelectric performance in bulk SrTiO_3 via nano-scale modulation doping. *Nano Energy* **35**, 387–395 (2017).
49. Ohtaki, M., Araki, K. & Yamamoto, K. High thermoelectric performance of dually doped ZnO ceramics. *J. Electron. Mater.* **38**, 1234–1238 (2009).
50. Ahmad, A. *et al.* Thermoelectric Performance Enhancement of Vanadium Doped n-Type In_2O_3 Ceramics via Carrier Engineering and Phonon Suppression. *ACS Appl. Energy Mater.* **3**, 1552–1558 (2020).
51. Bocher, L. *et al.* $\text{CaMn}_{1-x}\text{Nb}_x\text{O}_3$ ($x \leq 0.08$) perovskite-type phases as promising new high-temperature n-type thermoelectric materials. *Inorg. Chem.* **47**, 8077–8085 (2008).
52. Liu, H. *et al.* Enhanced thermoelectric properties of nonstoichiometric $\text{TiO}_{1.76}$

- with excellent mechanical properties induced by optimizing processing parameters. *Ceram. Int.* **44**, 19859–19865 (2018).
53. Tan, X. *et al.* Synergistical Enhancement of Thermoelectric Properties in n-Type Bi₂O₂Se by Carrier Engineering and Hierarchical Microstructure. *Adv. Energy Mater.* **9**, 1–7 (2019).
 54. Zhang, L., Lin, D. Y., Wang, H., Car, R. & Weinan, E. Active learning of uniformly accurate interatomic potentials for materials simulation. *Phys. Rev. Mater.* **3**, 023804 (2019).
 55. Zhang, Y. *et al.* DP-GEN: A concurrent learning platform for the generation of reliable deep learning based potential energy models. *Comput. Phys. Commun.* **253**, 107206 (2020).
 56. He, R. *et al.* Ferroelastic Twin-Wall-Mediated Ferroelectriclike Behavior and Bulk Photovoltaic Effect in SrTiO₃. *Phys. Rev. Lett.* **132**, 1–7 (2024).
 57. He, R. *et al.* Structural phase transitions in SrTiO₃ from deep potential molecular dynamics. *Phys. Rev. B* **105**, 1–10 (2022).
 58. Tadano, T. & Tsuneyuki, S. Self-consistent phonon calculations of lattice dynamical properties in cubic SrTiO_3 with first-principles anharmonic force constants. *Phys. Rev. B* **92**, 054301 (2015).
 59. Togo, A. First-principles Phonon Calculations with Phonopy and Phono3py. *J. Phys. Soc. Japan* **92**, (2023).
 60. Togo, A., Chaput, L., Tadano, T. & Tanaka, I. Implementation strategies in

phonopy and phono3py. *J. Phys. Condens. Matter* **35**, 353001 (2023).

61. Deringer, V. L., Tchougréeff, A. L. & Dronskowski, R. Crystal orbital Hamilton population (COHP) analysis as projected from plane-wave basis sets. *J. Phys. Chem. A* **115**, 5461–5466 (2011).
62. Xiao, Y. *et al.* Band Sharpening and Band Alignment Enable High Quality Factor to Enhance Thermoelectric Performance in n-Type PbS. *J. Am. Chem. Soc.* **142**, 4051–4060 (2020).
63. He, W. *et al.* High thermoelectric performance in low-cost SnS_{0.91}Se_{0.09} crystals. *Science* (80-.). **365**, 1418–1424 (2019).

REVIEWERS' COMMENTS

Reviewer #1 (Remarks to the Author):

Authors well addressed all my concerns, it can be published as is.

Reviewer #2 (Remarks to the Author):

Authors appropriately reply to my comments and revised manuscript. So I think it can be published in Nature Communications.

Reviewer #3 (Remarks to the Author):

Dera Aditor,

I believe the authors have addressed and responded to all the questions raised by the reviewers.

The paper is ready for acceptance as it stands.

Best regards

Reviewer #4 (Remarks to the Author):

Response to Reviewers

Reviewer #1 (Remarks to the Author):

Authors well addressed all my concerns, it can be published as is.

Response:

Thanks very much for the reviewer's kind suggestions on our manuscript.

Reviewer #2 (Remarks to the Author):

Authors appropriately reply to my comments and revised manuscript. So I think it can be published in *Nature Communications*.

Response:

Thanks very much for the reviewer's recognition and valuable suggestions on our manuscript.

Reviewer #3 (Remarks to the Author):

I believe the authors have addressed and responded to all the questions raised by the reviewers.

The paper is ready for acceptance as it stands.

Response:

Happy to hear that our answers satisfied the reviewer. Thanks very much for the reviewer's recognition and valuable suggestions on our manuscript.

Reviewer #4 (Remarks to the Author):

Response:

Thanks for the reviewer's dedication to this work.